# ON THE THEORY OF CONTINUAL LEARNING WITH GRADIENT DESCENT FOR NEURAL NETWORKS

## ABSTRACT

Continual learning, the ability of a model to adapt to an ongoing sequence of tasks without forgetting the earlier ones, is a central goal of artificial intelligence. To shed light on its underlying mechanisms, we analyze the limitations of continual learning in a tractable yet representative setting. In particular, we study one-hidden-layer quadratic neural networks trained by gradient descent on an XOR cluster dataset with Gaussian noise, where different tasks correspond to different clusters with orthogonal means. Our results obtain bounds on the rate of forgetting during train and test-time in terms of the number of iterations, the sample size, the number of tasks, and the hidden-layer size. Our results reveal interesting phenomena on the role of different problem parameters in the rate of forgetting. Numerical experiments across diverse setups confirm our results, demonstrating their validity beyond the analyzed settings.

## 1 INTRODUCTION

### 1.1 MOTIVATION

Gradient-based methods are the primary approach for training neural networks. In recent years, research in learning theory has shown that neural networks can efficiently learn various data classes using empirical risk minimization (ERM) methods. In many real-world settings, data arrive sequentially in a non-stationary manner, requiring the learner to maintain performance on past tasks while acquiring new capabilities. In such cases, a learning model must be continually learnable, meaning it should retain previously acquired knowledge when trained on new tasks. On the other hand, various learning systems, including deep learning architectures, can be prone to *catastrophic forgetting*, that is, updating a model on new data causes a dramatic drop in performance on previously learned tasks (McCloskey & Cohen, 1989; Goodfellow et al., 2013). The goal of continual (lifelong) learning is to develop models and methods that, even without retraining on old data, experience minimal forgetting when incorporating new information.

Despite deep learning's ubiquity, characterizing the power and limitations of neural networks is still an ongoing research direction. While several recent works aim to understand the power of gradient descent (GD) for training neural networks with stylized data distributions, these works are still limited to single-task scenarios (for some examples see (Du et al., 2019; Bartlett et al., 2021; Abbe et al., 2022)). However, the strengths and limitations of gradient descent in continual learning remain largely unexplored.

In this work, we present several results on the performance of gradient descent in neural networks for scenarios where there is a stream of independent tasks on which the model is sequentially trained. We mainly focus on studying unregularized ERM for this problem and identify regimes and conditions for clustered synthetic datasets where gradient descent without any explicit regularization is capable of achieving arbitrarily small forgetting and small test error for all tasks simultaneously. In doing so, we consider a simple but illustrative nonlinear data distribution for multiple independent tasks based on XOR clusters, and characterize the sample, iteration, and computation complexities based on data dimension and number of tasks for successful continual learning. We are also able to characterize the forgetting error in terms of problem variables for a given task after training the network on arbitrary number of subsequent tasks. We show that both train- and test-time forgetting errors can be mitigated by increasing the sample size of the subsequent tasks.

**Techniques and Contributions.** Our method is based on the decomposition of the test-time forgetting error into two terms based on forgetting in training loss and the generalization gap caused by intermediate learning tasks. We bound the generalization gap by an argument based on algorithmic stability (Bousquet & Elisseeff, 2002; Lei & Ying, 2020a) tailored to our set-up of continual learning with neural nets, which leads to conditions on the network width and the number of iterations and samples to achieve a small generalization error after learning independent intermediate tasks from distinct distributions. We then use a data-specific argument to formulate the evolution of the learned weights throughout the gradient descent steps to bound the training loss and forgetting. In particular, we first consider an asymptotic regime where for both the sample size and network size $m, n \to \infty$. The critical observation here is that in this regime, for every task, the gradient at initialization is in the correct direction, and with sufficient number of GD steps, the train loss and the amount of forgetting(i.e., the increase in training loss caused by learning later tasks) are asymptotically zero. As a result of this and with concentration bounds for finite $n$ and $m$, we are able to characterize the rate of forgetting based on these parameters. To the best of our knowledge, our results are the first closed-form guarantees for the train and test performance of continual learning methods when using neural networks and they predict several of the empirical observations on the role of training-set size and over-parameterization. We summarize our contributions in the following:

- We prove bounds for forgetting in continual learning using neural nets, showing for the $d$-dimensional XOR cluster dataset that train-time forgetting after training for $K$ subsequent tasks is bounded by $\widetilde{O}(\eta T \frac{\sqrt{K}}{d\sqrt{n}} + \eta T \frac{\sqrt{K}}{d^2 \operatorname{poly} \log(d)} + \eta^2 T^2 \frac{K^2}{\sqrt{m}})$ where $T$ and $n$ denote the number of GD iterations and sample size for each subsequent task, respectively, and $m$ denotes the hidden layer size.

- We characterize the sample and computation complexity for continual learning, derive a rate of $n = \widetilde{\Theta}(d^2 K), m = \Theta(d^8 K^4), T = \widetilde{\Theta}(d^2)$ for the number of samples, hidden-layer size, and number of GD iterations, respectively, to achieve small training loss for all $K$ tasks.

- We derive a bound on the test-time forgetting by decomposing it into the train-time forgetting term and a delayed generalization term, and we show the above complexities also lead to vanishing test-time forgetting.

- Numerical experiments support our theoretical insights across diverse problem settings, demonstrating their applicability to models and data distributions beyond those explicitly analyzed.

## 1.2 PRIOR WORKS

The main algorithms for continual learning are based on functional (or architectural) regularization (Li & Hoiem, 2017; Kirkpatrick et al., 2017; Sharif Razavian et al., 2014) or experience replaying (Schaul et al., 2016; Rolnick et al., 2019). In order to mitigate forgetting, regularization-based methods enforce the new solutions to remain close to solutions to previous tasks. On the other hand, it has been hypothesized that the network width has a similar impact (Graldi et al., 2024), since increasing the width enables the network to operate in the lazy/kernel regime where it is known that the network's weights do not travel a significant distance from their initialization point and the features remain constant during training (Jacot et al., 2018; Ghorbani et al., 2019). It is therefore natural to ask to what extent the width helps continual learning. Few works have dealt with this question. In particular, the impact of the width of the network on continual learning was studied in (Guha & Lakshman, 2024; Graldi et al., 2024; Mirzadeh et al., 2022a;b; Wenger et al., 2023). For instance, (Mirzadeh et al., 2022a;b) observed the impact of width in improving catastrophic forgetting and noticed that increasing the width always mitigates forgetting. However, (Wenger et al., 2023) claimed that such improvements vanish when the network is trained for a sufficiently large number of iterations until convergence. More recently, (Graldi et al., 2024) attempted to resolve the issue claiming that improvements only happen in the kernel regime, where there is early stopping to avoid weights moving a significant distance from their initialization. Our theoretical and empirical results on the impact width for the XOR cluster data also verify the benefits of width in the kernel regime with early stopping.

(Guha & Lakshman, 2024) showed analytically through a general argument that increasing the width helps continual learning, although the improvements shrink as width grows. The dependence on width in their bound is not explicitly determined, and moreover, the bound does not depend on the underlying algorithm or number of samples. In contrast, our analysis is algorithm-dependent

and yields closed-form bounds, explicitly highlighting the roles of different problem parameters in test-time forgetting.

Perhaps the closest works to ours are (Doan et al., 2021; Bennani et al., 2020; Lee et al., 2021; Karakida & Akaho, 2021), which derived general expressions to characterize forgetting in neural networks in the lazy regime. A recent work by (Li et al., 2025) focuses specifically on CNNs in a multi-view data model and characterizes forgetting. (Benjamin et al., 2024) approach uses an "ensemble/NTK" perspective treating networks in the lazy regime and gives a reinterpretation of continual learning. (Andle & Yasaei Sekeh, 2022) focus on layer-wise information flow and develop a probabilistic theory for CL performance across layers. However, these results do not lead to closed-form bounds and are applicable for different models such as CNNs, while our results yield the first bounds for a multi-index model learned by neural nets.

Another related line of work has focused on linear classification/regression in the realizable regime, where a single linear solution can interpolate data from all tasks (Goldfarb & Hand, 2023; Lin et al., 2023; Evron et al., 2023; Banayeeanzade et al., 2024). In particular, (Evron et al., 2023) analyzed catastrophic forgetting through the lens of implicit bias in linear classification across various setups, including cyclic and random task orderings. (Cao et al., 2022) derived sample complexity of continually learning linear models and GLMs. In contrast, we adopt a more practical perspective by examining sample complexity, early stopping, and the effects of over-parameterization in a stylized neural network setting.

Our analysis of the generalization error is done through the lens of the algorithmic-stability framework and follows the approach in (Hardt et al., 2016; Feldman & Vondrak, 2019; Lei & Ying, 2020a;b; Richards & Kuzborskij, 2021; Taheri & Thrampoulidis, 2024), extending it to accommodate the continual learning setting. Our results reveal that generalization gap for continual learning is impacted by the training loss of later tasks (as in Thm 4) or number of tasks (as in Thm 3) which is new compared to single-task analyses. For the training-loss analysis (Thm 1-2), we use a new approach based on a double-asymptotic regime where first we consider the regime of $m \to \infty$ in order to characterize the weights for any number of iterations and then consider the asymptotes of $n \to \infty$ in order to characterize the role of number of samples on the train-time forgetting. The final bound is obtained by deriving concentration error of finite-width networks for every GD iteration. This differs from the existing analyses of neural nets for single-task classification setups in the lazy regime which are mainly based on class margin (Nitanda et al., 2019; Ji & Telgarsky, 2020; Taheri & Thrampoulidis, 2024).

**Notation.** We use the standard complexity notation $\lesssim, o(\cdot), O(\cdot), \Theta(\cdot), \Omega(\cdot)$ and denote $\widetilde{o}(\cdot), \widetilde{O}(\cdot), \widetilde{\Theta}(\cdot), \widetilde{\Omega}(\cdot)$ to hide poly-logarithmic factors. The subscripts in $O_d(\cdot), o_d(\cdot)$ denote the dependence on the parameter $d$. We use $\|\cdot\|$ for the $\ell_2$ norm of vectors. We denote $[n] := \{1, 2, \cdots, n\}$. The expectation and probability with respect to the randomness in $\mathcal{D}$ are denoted by $\mathbb{E}_{\mathcal{D}}[\cdot], \mathrm{Pr}_{\mathcal{D}}(\cdot)$. The gradient of the model $\Phi : \mathbb{R}^{p \times d} \to \mathbb{R}$ with respect to the first input (weights) is denoted by $\nabla \Phi$.

## 2 MAIN RESULTS

### 2.1 PROBLEM SETUP

#### 2.1.1 GRADIENT-BASED CONTINUAL LEARNING WITH NEURAL NETWORKS

We consider the problem of sequentially learning $K$ independent tasks, where each task is trained in isolation but in a fixed order. Specifically, for the $k$-th task, we perform $T$ iterations of gradient descent using a dataset of $n$ training samples. The objective of task $k$ is defined as

$$\widehat{F}(w, \mathcal{D}_k) = \frac{1}{n} \sum_{i=1}^{n} f\big(y_i \, \Phi(w, x_i)\big),$$

where $\mathcal{D}_k = \{(x_i, y_i)\}_{i=1}^{n}$ denotes the set of training examples for task $k$, and the mapping $\Phi$ represents a two-layer neural network with $m$ hidden neurons and activation $\phi$, given by

$$\Phi(w, x) = \frac{1}{\sqrt{m}} \sum_{i=1}^{m} a_i \, \phi(x^\top w_i).$$

---

**Algorithm 1:** Continual Learning with Gradient Descent

---

**Input:** Number of tasks $K$, number of steps per task $T$, learning rate $\eta$

**Output:** Final model parameters $w_K$

1 Initialize model parameters $w_1^{(0)} \sim \mathcal{N}(0, I_p)$;

2 **for** $k = 1$ **to** $K$ **do**

3      Load task-specific dataset $\mathcal{D}_k$;

4      **for** $t = 0$ **to** $T - 1$ **do**

5          Sample mini-batch(or full-batch) $\mathcal{B}_t \subseteq \mathcal{D}_k$;

6          $w_k^{(t+1)} \leftarrow w_k^{(t)} - \eta \nabla \widehat{F}(w_k^{(t)}; \mathcal{B}_t)$;

7      Set $w_{k+1}^{(0)} \leftarrow w_k := w_k^{(T)}$;

8 **return** $w_K := w_K^{(T)}$

---

Throughout the paper, we assume that the output layer coefficients $a_i \in \{\pm 1\}$ are fixed, let $f$ be the hinge-loss and we focus on the case of quadratic activation where $\phi(t) = t^2/2$. For convenience, we denote the empirical loss for task $k$ by $\widehat{F}_k(w) := \widehat{F}(w, \mathcal{D}_k)$, and the corresponding population (test) loss by $F_k(w) := F(w, \bar{\mathcal{D}}_k) = \mathbb{E}_{(x,y) \sim \bar{\mathcal{D}}_k}\big[f\big(y\,\Phi(w, x)\big)\big]$, where the expectation is taken over the test-set distribution $\bar{\mathcal{D}}_k$.

The complete continual learning procedure is summarized in Algorithm 1. We initialize the parameter vector $w_1^{(0)}$ from a standard Gaussian distribution, $w_1^{(0)} \sim \mathcal{N}(0, I_p)$, where $p = md$ is the total number of trainable parameters in the first layer. For each task $k \in \{1, \ldots, K\}$, we train the network starting from initialization $w_{k-1}^{(T)}$ for $T$ gradient descent updates on $\widehat{F}_k$. The resulting vector after finishing the training on task $k$ is denoted by $w_k := w_k^{(T)} := w_{k+1}^{(0)}$, and it serves as the initialization for the subsequent task $k+1$. After processing all $K$ tasks, the algorithm outputs the final parameter vector $w_K$, which contains the accumulated knowledge obtained from the entire sequence of tasks.

### 2.1.2 XOR CLUSTER DATASET

Consider data according to the XOR cluster distribution with Gaussian noise where $x \in \mathbb{R}^d, y \in \{\pm 1\}$ and

$$x \sim \begin{cases} \frac{1}{2}\mathcal{N}(\mu_+, \sigma^2 I_d) + \frac{1}{2}\mathcal{N}(-\mu_+, \sigma^2 I_d) & \text{if } y = 1, \\ \frac{1}{2}\mathcal{N}(\mu_-, \sigma^2 I_d) + \frac{1}{2}\mathcal{N}(-\mu_-, \sigma^2 I_d) & \text{if } y = -1, \end{cases} \tag{1}$$

where $\mu_+ \perp \mu_-$, and $\Pr[y = 1] = \Pr[y = -1] = 1/2$. This dataset serves as a representative example of a realizable, not linearly separable problem that is well-suited for analyzing neural networks. The XOR cluster and its Boolean variant (known as parities) have been extensively studied in the deep learning theory literature (Wei et al., 2019; Refinetti et al., 2021; Xu et al., 2024; Telgarsky, 2023; Taheri & Thrampoulidis, 2024; Glasgow, 2024; Taheri et al., 2025). In particular, the XOR model is a representative instance of multi-index models, which have recently been used to investigate the sample complexity of neural network learning(Damian et al., 2022; Ba et al., 2022; Abbe et al., 2022). For this data set, we show that $d^2$ samples and $d^4$ neurons are sufficient to achieve near zero train and test loss (see Prop. 2 in App. A).

For the continual learning setup we consider a stream of $K$ tasks, where each task is generated according to the XOR cluster dataset, that is, for task $k$:

$$x \sim \begin{cases} \frac{1}{2}\mathcal{N}(\mu_+^k, \sigma^2 I_d) + \frac{1}{2}\mathcal{N}(-\mu_+^k, \sigma^2 I_d) & \text{if } y = 1, \\ \frac{1}{2}\mathcal{N}(\mu_-^k, \sigma^2 I_d) + \frac{1}{2}\mathcal{N}(-\mu_-^k, \sigma^2 I_d) & \text{if } y = -1. \end{cases} \tag{2}$$

We assume that $\mu_+^k$ and $\mu_-^k$ are mutually orthogonal for all $k \in [K]$, with $\|\mu_+^k\| = \|\mu_-^k\| = \Theta(\frac{1}{\sqrt{d}})$, $\Pr[y = 1] = \Pr[y = -1] = 1/2$, and noise level $\sigma = \Theta(\frac{1}{\log^c(d)\sqrt{d}})$ for some universal constant $c$. The orthogonality assumption reflects the fact that tasks are not correlated. Although our analysis can be extended to the more general case where the mean vectors are not orthogonal between tasks,

this is beyond the scope of the present work. We further assume that the number of tasks grows at most poly-logarithmically with the data dimension, i.e., $K = \widetilde{O}_d(1)$.

**Forgetting and Continual Learning.** Let $w_k$ denote the weights after training with data from task $k$ for some $k \in [K]$. *Test-time forgetting* is measured by the increase in test loss for the $k$th task after training on $K - k$ subsequent tasks:

$$\text{Test-time Forgetting: } \mathcal{F}^{\text{ts}}_{k,K} := F_k(w_K) - F_k(w_k).$$

We can decompose the test-time forgetting as follows:

$$\mathcal{F}^{\text{ts}}_{k,K} = [F_k(w_K) - \widehat{F}_k(w_K)] + [\widehat{F}_k(w_K) - \widehat{F}_k(w_k)] + [\widehat{F}_k(w_k) - F_k(w_k)].$$

In the interpolating regime where the network can achieve zero training loss, we can drop the last term and bound the test-time forgetting based on *generalization gap* and *training loss*:

$$\mathcal{F}^{\text{ts}}_{k,K} \leq \underbrace{\left[\widehat{F}_k(w_K) - \widehat{F}_k(w_k)\right]}_{\text{Train-time forgetting } \mathcal{F}^{\text{tr}}_{k,K}} + \underbrace{\left[F_k(w_K) - \widehat{F}_k(w_K)\right]}_{\text{Delayed generalization gap}}. \quad (3)$$

In the following section, we discuss each term separately. When combined, these will give an upper bound on the expected test-time forgetting.

## 2.2 TRAIN AND TEST-TIME FORGETTING BOUNDS

The following theorem provides closed-form bounds on the train-time forgetting of task $k$ after learning the subsequent $K - k$ tasks (for a total of $K$ tasks). We assume the hinge loss, $f(u) = \max\{1 - u, 0\}$, and adopt the data distribution specified in Eq. 2. The proofs for the theorems in this section are deferred to the appendix.

**Theorem 1** (Train-time forgetting). *Consider the $d$-dimensional XOR cluster dataset with $K$ tasks and assume gradient descent with $\eta T = \Theta(d^2)$ iterations and $n = \widetilde{\Theta}(d^2 K)$ samples for each subsequent task trained by a neural net with $m = \widetilde{\Omega}(d^8 K^4)$ hidden neurons. Then, with high probability, the train-time forgetting is $\mathcal{F}^{\text{tr}}_{k,K} = o_d(1)$. In particular, with probability $1 - \delta$, we have:*

$$|\mathcal{F}^{\text{tr}}_{k,K}| := |\widehat{F}_k(w_K) - \widehat{F}_k(w_k)| = \widetilde{O}\left(\eta T \frac{\sqrt{K-k}}{d\sqrt{n}} + \eta T \frac{\sqrt{K-k}}{d^2 \operatorname{poly}\log(d)} + \eta^2 T^2 \frac{K^2}{\sqrt{m}}\right), \quad (4)$$

*where $\widetilde{O}(\cdot)$ hides logarithmic factors in $n, T$ and $\delta$.*

The first and third terms in Eq. 4 capture the effects of sample size and hidden-layer width. Importantly, neither factor alone is sufficient to eliminate train-time forgetting. However, with sufficiently large $n$ and $m$, we obtain a forgetting rate $\mathcal{F}^{\text{tr}}_{k,K} = O(1/\operatorname{poly}\log(d)) = o_d(1)$. Here, $n$ denotes the sample size of datasets learned after task $k$. Although these subsequent tasks are independent of and orthogonal to task $k$ (tasks are IID with orthogonal means), their larger training sets nevertheless enhance the overall continual learning process. Our experiments in Section 3, conducted across different activation functions, loss functions, and datasets under various problem settings, empirically confirm the theoretical roles of network width, sample size, and the number of tasks.

We note that the early-stopping choice $\eta T = \widetilde{\Theta}(n)$ is standard in the deep learning literature, particularly in the interpolation regime for single-task settings (Ji & Telgarsky, 2020; Lei & Ying, 2020a), as it ensures the training loss is driven close to zero. As the following theorem demonstrates, under this choice the training loss remains uniformly small across all tasks.

**Theorem 2** (Train error in continual learning). *Let the assumptions of Theorem 1 hold. Then, after $KT$ iterations of GD, with high probability, the misclassification train error and train loss are $o_d(1)$ uniformly for all $K$ tasks.*

The proofs of Theorems 1-2 are deferred to App. C. A combination of these theorems yields sufficient conditions for successful continual learning as measured by training performance. We remark that the proof of both theorems, up to calculations related to the model-output's equations in Eq.

14 or forgetting equation in Eq. 16 , hold for a broad family of data distributions. These steps require no special structure beyond concentration of the empirical NTK and uniform boundedness of inputs. The parts of the analysis that specialize to the XOR-cluster distribution primarily arise when deriving explicit closed-form expressions for the model output and for characterizing closed-form bounds for forgetting. Extending the bounds to more general data distributions (e.g. other clustered multi-index models) would therefore require replacing the concentration steps and explicit calculations with distribution-specific estimates. We expect the qualitative dependencies on $(n, m, T, \eta, K)$ to remain similar under other clustered or sub-Gaussian task distributions, but the exact forms will change.

Our next result derives the delayed generalization gap (as defined in Eq. 3) for almost any data distribution. In fact, it also shows that the above assumptions for $m, n$ and $T$ are also sufficient for good continual *test-time* performance for the XOR cluster dataset.

**Theorem 3** (Delayed generalization gap)**.** *Assume the loss function is 1-Lipschitz and 1-smooth. Then, the expected delayed generalization gap satisfies,*

$$\mathcal{F}_{k,K}^{\text{gen}} := \mathbb{E}_{\mathcal{D}_k}\left[F_k(w_K) - \widehat{F}_k(w_K)\right] \lesssim \frac{\eta T\, e^{\frac{\eta T(K-k+1)}{\sqrt{m}}}}{n}.$$

**Remark 1** (Test-time forgetting)**.** *Note that the gap decays with the rate $1/n$ and similar to the train-time forgetting, given sufficiently large width, it is linearly proportional to the number of iterations. While the dependence on $T$ is unfavorable, and in general we expect a time-independent generalization gap, we note that with the training loss guarantees from Theorems 1-2, and in view of Theorem 3 we find that with $n = \widetilde{\Theta}(d^2 K) = \widetilde{\Theta}(\eta T)$ samples and with $m = \widetilde{\Omega}(d^8 K^4)$, it holds $\mathcal{F}_{k,K}^{\text{gen}} = o_d(1)$ resulting in vanishing test-time forgetting in view of Eq. 3. Finally, we note -as the proof shows- training occurs within the linear region of the hinge loss, which allows us to combine the results of the previous theorems despite the smoothness assumption on the loss in Theorem 3.*

Under additional conditions on continual learnability of each task and a self-bounded assumption for the loss function(i.e., $|f'(u)| < f(u)$) that includes logistic loss $f(u) = \log(1 + \exp(-u))$, in the next theorem, we prove a tighter generalization bound, which has a noticeably milder dependence on $T$ compared to Theorem 3.

**Theorem 4** (Improved gen. gap)**.** *Assume the loss function is self-bounded, 1-Lipschitz and 1-smooth. Let the network's width $m$ be large enough so that $\sqrt{m} \gtrsim \eta \sum_{j=k+1}^{K} \sum_{t=0}^{T-1} \widehat{F}_j(w_j^{(t)})$. Moreover, assume there exists $w_k^\star$ achieving small training loss $\widehat{F}_k(w_k^\star) \leq \|w_k^\star - w_k^{(0)}\|^2/(\eta T)$ for task $k$, and satisfying $m \gtrsim \|w_k^\star - w_k^{(0)}\|^4$. Then,*

$$\mathcal{F}_{k,K}^{\text{gen}} \lesssim \frac{\eta}{n}\mathbb{E}_{\mathcal{D}_k}\left[e^{\frac{\eta}{\sqrt{m}}c_{k,K}} \sum_{t=0}^{T-1} \widehat{F}_k(w_k^{(t)})\right], \tag{5}$$

*where $c_{k,K} = O\left(\sum_{j=k+1}^{K} \sum_{t=0}^{T-1} \widehat{F}_j(w_j^{(t)})\right)$.*

As the result shows, $\mathcal{F}_{k,K}^{\text{gen}}$ decays with both the cumulative training loss of the later tasks as in $c_{k,K}$ and the network width, and it is proportional to the cumulative training loss of task $k$. In particular, the cumulative training loss can be much smaller than $T$, potentially leading to tighter bounds compared to the results of previous theorem.

**Remark 2.** *In words, the conditions on $\|w_k^\star - w_k^{(0)}\|$ ensure that task $k$ remains learnable in the kernel regime, i.e., the initialization is sufficiently close to the task-specific optimum so that optimization can succeed. To better interpret this result, let us consider the case where $k = 1$, and we are interested in bounds on $\mathcal{F}_{1,K}^{\text{gen}}$ for some $K \geq 2$. First, we note that for the XOR cluster dataset, there exists(see Proposition 2 in App. A) $w_1^\star$ such that $\|w_1^\star - w_1^{(0)}\| = \Theta(d \cdot \log(T))$ and $\widehat{F}_1(w_1^\star) \leq \frac{1}{T}$, leading to train-loss $\widehat{F}_1(w_1^{(t)}) = O(\frac{d^2 \log^2(t)}{t})$. Therefore, in view of Theorem 4, if $\sqrt{m} \gtrsim \eta \sum_{j=2}^{K} \sum_{t=0}^{T-1} \widehat{F}_j(w_j^{(t)})$ and $m \gtrsim d^4 \log^4(T)$, the expected generalization gap after $T$ iterations for each of $K$ tasks satisfies,*

$$\mathcal{F}_{1,K}^{\text{gen}} \lesssim \frac{\eta d^2 \log^3(T)}{n},$$

*where we can hide the exponential term in Eq. 5 for simplicity since the exponent is constant under the condition on $m$. This shows that Theorem 4 may lead to bounds with significantly better dependence based on $T$ compared to Theorem 3 (poly-logarithmic vs linear). Although this result cannot be combined directly with our setting for the training loss (since the hinge loss considered for the training-loss analysis is not self-bounded) it still provides valuable insight. In particular, it can be interpreted as a stronger extension of Theorem 3, highlighting how the training loss directly influences the generalization gap in continual learning as shown by Eq. 5.*

## 2.3 REGULARIZED CONTINUAL LEARNING

We consider the regularized continual learning algorithm (Aljundi et al., 2017; Kirkpatrick et al., 2017; Lewkowycz & Gur-Ari, 2020) with parameter $\lambda$ where for each task $k \geq 2$, the objective is to minimize the following,

$$\min_w \ \widehat{F}_k(w) + \frac{\lambda}{2}\|w - w_{k-1}\|^2. \tag{6}$$

The regularization parameter $\lambda$ can be chosen to be fixed, time-varying or data-dependent (Evron et al., 2023; Lewkowycz & Gur-Ari, 2020; Kirkpatrick et al., 2017). In the next proposition, we consider the fixed $\lambda$ in order to study the effects of regularization on the GD iterates. We show that in the linearized regime (i.e., the infinite-width regime) where the network output can be written as a first-order approximation around initialization, the regularized continual learning problem is effectively equivalent to unregularized minimization with a time-varying step-size.

**Proposition 1** (Regularized continual learning)**.** *Consider the regularized continual learning problem Eq.6 in the linearized regime, with the same setup as Theorem 1. The iterates of this algorithm with step-size $\eta$ are equivalent to unregularized continual learning with step-size $\widetilde{\eta}_T$ for any task $k \geq 2$, where we define $\widetilde{\eta}_T := \frac{\alpha_T \eta}{T}$ and $\alpha_T := \frac{1-(1-\eta\lambda)^T}{\eta\lambda}$.*

Hence, as $T$ increases, the effective step-size decreases, preventing iterations from moving a significant distance from the solution of previous task. The above result shows that in our setup with kernel regime and early stopping, regularized continual learning is equivalent to the unregularized one with a different step-size, implying that regularization cannot improve the results of the previous section.

## 3 EXPERIMENTS

We demonstrate the impact of sample size, number of tasks, and network width on the performance of continual learning for different loss functions, activations functions, data distributions, architectures, step-sizes and training horizons. We include the implementation details for each figure and additional experiments, including experiments on the MNIST dataset as well as transformer architecture, in Appendix E.

**Impact of sample-size, training horizon and number of Tasks**   The first data model we consider is the XOR cluster (Section 2.1) with orthogonal mean vectors. Figure 1 shows how sample-size affects the train-loss forgetting for $K = 3$ tasks using quadratic activation and linear loss. Here, we increase the sample size for each task from $n = 2500$ to $n = 5000$, showing how the increase can diminish test-error forgetting. Figure 8 in the appendix repeats this experiment for different problem parameters. The observations from both plots are in-line with our theoretical insights on the role of sample-size on train and test time forgetting.

In order to verify the role of sample size of later tasks on train-time forgetting, we consider an experiment where the sample-size for task 1 is fixed, and for later tasks we increase the sample-size. The resulting training loss curves for different loss functions and activations are shown in Figures 2,3, and 9 (in the appendix). In accordance with Theorem 1, it can be observed that increasing the sample-size on tasks 2,3 has a positive influence on the forgetting of task 1. This implies that increasing the sample-size not only stabilizes the per-task training loss, but also reduces the amount of forgetting for previous tasks. While we use linear loss with quadratic activation for Figure 2, Figures 3, 9 indicate these observations extend to different losses and activations including the commonly used logistic loss and the ReLU and GELU activations.

In Figure 4, we consider $K = 6$ tasks of the XOR cluster dataset and increase $T$ from $T = 2000$ to $T = 4000$ for each task with $n = 200, 800, 2000$ samples per each task. Note that increasing $T$, deteriorates the training loss for task 1 as training progresses. While increasing $T$ helps with training loss for task 1 at the end of training of task 1, (the dashed lines are below the solid lines at $k = 1$ for any value of $n$), the amount of increase in the training loss for $T = 4000$ is larger than $T = 2000$, eventually leading to larger training loss for task 1 as $K$ increases. The right panel in Figure 4 shows the training loss for each task during learning these 6 tasks, illustrating that the train loss achieves near zero training loss for each task. On the other hand, increasing $n$ for each task, helps with diminishing the training loss. To better see this impact, in Figure 10 in the appendix, we increase the number of tasks and consider learning $K = 15$ and $K = 20$ tasks of the XOR cluster dataset. These plots again verify our insights on the role of training-set size. The impact of increasing tasks is also visible in the Left figure while using GELU activation and the logistic loss.

**Impact of over-parameterization.** In Figure 5 we consider the XOR cluster dataset for $K = 3$ tasks with Quadratic activation and gradually increase $m$ from $m = 10^2$ to $m = 10^4$. We find that

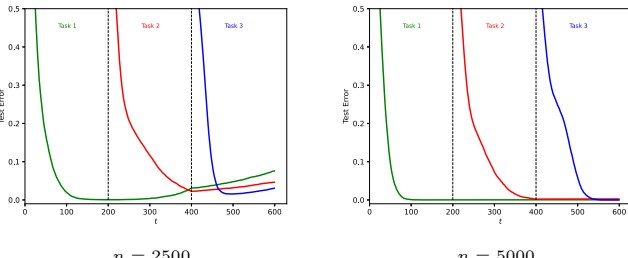

$$n = 2500 \qquad\qquad n = 5000$$

Figure 1: Classification test error for each task vs iterations for the XOR cluster with $K = 3$ tasks trained on a quadratic network with $n = 2500$(left) and $n = 5000$(right) training samples per task.

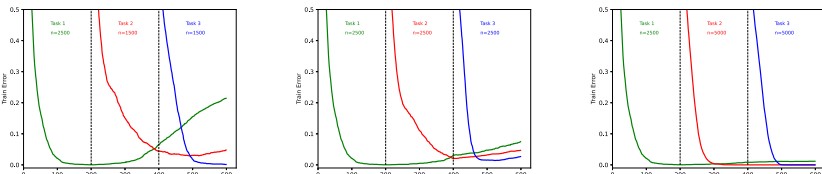

Figure 2: Classification train error for each task vs iterations for the XOR cluster with $K = 3$ tasks trained on a quadratic network. We fix $n = 2500$ for the first task and increase the sample size of second and third tasks across figures. Increasing the sample-size stabilizes per-task training and *decreases forgetting* for *previous tasks*.

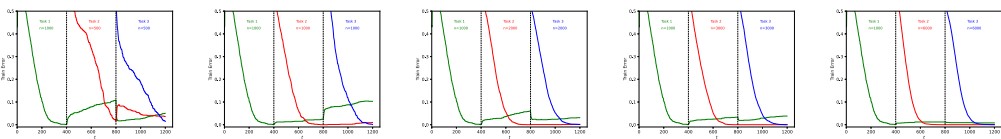

Figure 3: We repeat the experiment from Figure 2, this time using GELU activation and logistic loss function, demonstrating that our findings remain valid across different settings.

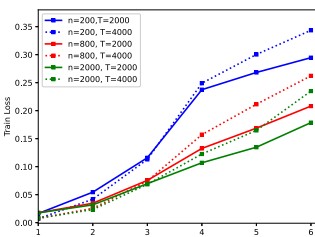 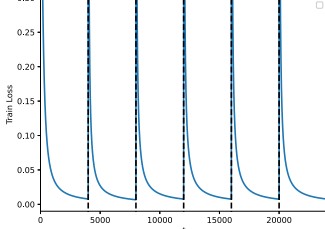

Figure 4: Left: Training loss of task 1 versus task index (i.e., $\widehat{F}_1(w_k)$ as a function of $k$) for $K = 6$ tasks for different sample-sizes and training horizons per task. Right: Training loss per task ($\widehat{F}_k(w_k^{(t)})$) versus iteration when $n = 2000, T = 4000$ for each task. We use GELU activation and logistic loss. While each task individually attains near-zero training loss, the training loss for the first task grows with both the number of tasks ($K$) and the number of iterations ($T$).

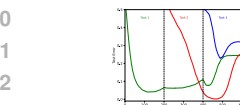 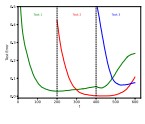 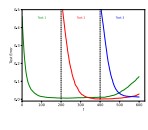 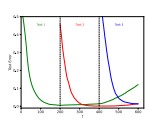 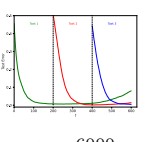 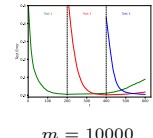

$m = 100$ $\quad\quad$ $m = 300$ $\quad\quad$ $m = 1000$ $\quad\quad$ $m = 3000$ $\quad\quad$ $m = 6000$ $\quad\quad$ $m = 10000$

Figure 5: Impact of network width ($m$) on the test error for learning the XOR cluster distribution with 3 tasks with quadratic networks. Increasing width helps with continual learning, however the benefits diminish as $m$ grows.

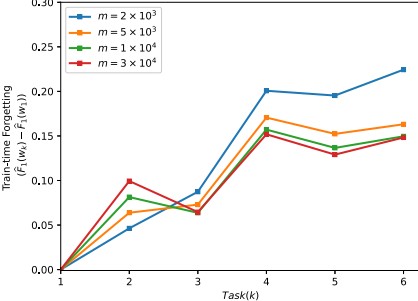

Figure 6: Train-time forgetting for task 1 vs task for $K = 6$ total tasks of the XOR cluster dataset for different over-parameterization choices. Here we use GELU activation and Logisitc loss and set $T = 10^3$ for each task.

increasing the width is generally beneficial for continual learning. However the benefits shrink as $m$ increases, where increasing the width from $m = 10^3$ to $m = 10^4$ has almost non-tangible impact on the overall performance of continual learning. Note that this is in line with Theorem 1, as we discussed the impact of width showing that width alone cannot reduce the train time forgetting to zero. We remark these insights also align with the *diminishing returns of width* phenomenon observed in previous works (Guha & Lakshman, 2024; Graldi et al., 2024) where the benefits of width decline as $m$ grows. In Figure 6, we consider learning $K = 6$ with the GELU activation and logistic loss for different choices of over-parameterization. The observations in this figure again verify our previous insights as increasing the width helps with continual learning, although it alone cannot lead to forget-less continual learning.

## 4 CONCLUSIONS AND FUTURE WORK

We studied gradient-based continual learning in a neural network setup, highlighting how different problem parameters affect catastrophic forgetting. Our analysis provides the first closed-form bounds on train and test time forgetting in this setting and clarifies the roles of sample size, width, number of tasks, and training horizon. There are several promising directions for future work. An immediate next step is to analyze other training methodologies, such as (mini-batch) stochastic gradient descent, where additional noise may interact with forgetting. Another important direction is to move beyond the quadratic two-layer setting and explore whether analogous guarantees can be obtained for richer architectures, including transformers. Our preliminary experiments in Figure 12 in the appendix show that some aspects of our results are observed, particularly for small transformers with Gaussian-Mixture data. Finally, our current analysis is limited to the lazy regime. Extending the theory to the feature-learning regime, where step-sizes are large, early stopping is avoided, and weights move significantly from their initialization, remains a challenging and exciting problem. While a recent work (Graldi et al., 2024) provides preliminary results on the drawbacks of feature learning for continual learning, more exploration in this regime could provide a more complete picture of continual learning in modern machine learning.

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
