APPENDIX

## A  SINGLE-TASK XOR CLUSTER

The next result derives the class margin for the single-task XOR cluster dataset and combined with standard results from the NTK literature it bounds the train and test loss for learning this dataset (as described by Eq. 1) with GD.

**Proposition 2** (Single-task XOR). *For the XOR cluster dataset for a given $T$, there exists target vector $w^\star \in \mathbb{R}^{dm}$ such that $\|w^\star - w_0\| = \Theta(d \cdot \log(T))$ and $\widehat{F}(w^\star) < 1/T$ and gradient descent with logistic loss and on a network with quadratic activation with width $m = \Omega(\|w^\star - w_0\|^4)$, achieves the training loss $\widehat{F}(w_t) = O(\frac{\|w^\star - w_0\|^2}{t})$ and the expected test loss $\mathbb{E}_{\mathcal{D}}[F(w_t)] = O(\frac{\|w^\star - w_0\|^2}{n})$ after $t = n$ GD iterations.*

*Proof.* Define four regions $R_1, R_2, R_3, R_4 \in \mathbb{R}^d$ such that

$$R_1 = \{x \in \mathbb{R}^d : x^\top(\mu_+ + \mu_-) > 0, x^\top(\mu_+ - \mu_-) > 0\},$$
$$R_2 = \{x \in \mathbb{R}^d : x^\top(\mu_+ + \mu_-) > 0, x^\top(\mu_+ - \mu_-) < 0\},$$
$$R_3 = \{x \in \mathbb{R}^d : x^\top(\mu_+ + \mu_-) < 0, x^\top(\mu_+ - \mu_-) > 0\},$$
$$R_4 = \{x \in \mathbb{R}^d : x^\top(\mu_+ + \mu_-) < 0, x^\top(\mu_+ - \mu_-) < 0\}.$$

Without loss of generality, assume $\mu_+ = [1/\sqrt{d}, 1/\sqrt{d}, 0, \cdots, 0]$ and $\mu_- = [-1/\sqrt{d}, 1/\sqrt{d}, 0 \cdots, 0]$. Our goal is to derive the NTK margin (Ji & Telgarsky, 2020; Taheri & Thrampoulidis, 2024) denoted by $\gamma$ for infinitely wide neural networks with initialization variable $z \in \mathbb{R}^d$, i.e., show that the equation below holds for all data points in the training set almost surely:

$$M(x_i, y_i) := y_i \int_{z \in \mathbb{R}^d} \phi'(\langle z, x_i \rangle) \langle w_z, x_i \rangle \, d\mu_{\mathcal{N}}(z) \geq \gamma$$

where $\mu_N$ is the standard Gaussian measure and $w_z$ is an initialization dependent vector such that $\|w_z\| \leq 1$ for all $z \in \mathbb{R}^d$. We drop the subscript $i$ and assume quadratic activation. Assume $y = 1, x \sim \mathcal{N}(\mu_+, \sigma I_d)$ without loss of generality. Then,

$$M = \int_{z \in R_1} \langle z, x \rangle \langle w_z, x \rangle \, d\mu_{\mathcal{N}}(z) + \int_{z \in R_2} \langle z, x \rangle \langle w_z, x \rangle \, d\mu_{\mathcal{N}}(z)$$
$$+ \int_{z \in R_3} \langle z, x \rangle \langle w_z, x \rangle \, d\mu_{\mathcal{N}}(z) + \int_{z \in R_4} \langle z, x \rangle \langle w_z, x \rangle \, d\mu_{\mathcal{N}}(z)$$

Let

$$w_z = \mu_+/\|\mu_+\|, -\mu_-/\|\mu_-\|, \mu_-/\|\mu_-\|, -\mu_+/\|\mu_+\| \text{ if } z \in R_1, R_2, R_3, R_4, \text{ respectively.}$$

Assume $s \sim \mathcal{N}(0, \sigma I_d)$.

$$\langle \mu_+/\|\mu_+\|, \mu_+ + s \rangle \int_{z \in R_1} \langle z, \mu_+ + s \rangle \, \mathrm{d}\mu_{\mathcal{N}}(z)$$

$$= (\|\mu_+\| + \mu_+^\top s/\|\mu_+\|) \left( (\frac{1}{\sqrt{d}} + s(1)) \int_{z \in R_1} z(1) d\mu_{\mathcal{N}}(z) + (\frac{1}{\sqrt{d}} + s(2)) \int_{z \in R_1} z(2) d\mu_{\mathcal{N}}(z) \right)$$

$$= (\|\mu_+\| + \frac{\mu_+^\top s}{\|\mu_+\|})(\frac{2}{\sqrt{d}} + s(1) + s(2))\mathbb{E}[z(1)\mathbf{1}_{z(1)>0}]$$

$$= \left( \sqrt{\frac{2}{d}} + \frac{\sqrt{2}}{2}(s(1) + s(2)) \right) \left( \frac{2}{\sqrt{d}} + s(1) + s(2) \right) \frac{1}{\sqrt{2}}$$

$$\gtrsim (\frac{1}{\sqrt{d}} + s(1) + s(2))^2$$

$$\gtrsim (\frac{1}{\sqrt{d}} + O(\frac{1}{\sqrt{d} \cdot \log^c(d)}))^2$$

$$\gtrsim 1/d.$$

For the second integral we have,

$$\langle -\mu_-/\|\mu_-\|, \mu_+ + s \rangle \int_{z \in R_2} \langle z, \mu_+ + s \rangle \ \mathrm{d}\mu_{\mathcal{N}}(z)$$

$$= \frac{-\mu_-^\top s}{\|\mu_-\|} \left( (\frac{1}{\sqrt{d}} + s(1)) \int_{z \in R_2} z(1) \ \mathrm{d}\mu_{\mathcal{N}}(z) + (\frac{1}{\sqrt{d}} + s(2)) \int_{z \in R_2} z(2) \ \mathrm{d}\mu_{\mathcal{N}}(z) \right)$$

$$= \frac{-1}{2}(s(2) - s(1))^2 = \Theta(\frac{1}{d \cdot \log^{2c}(d)})$$

For the third and fourth integral, due to symmetry, we reach the above final results again. Overall, we find that

$$M(x_i, y_i) \gtrsim \frac{1}{d} + O(\frac{1}{d \cdot \operatorname{poly} \log(d)}) = \Omega(1/d).$$

For data points coming from other three clusters of the XOR distribution, we reach the same conclusion. Therefore the margin scales as $1/d$ for every training sample. Using this margin result in (Taheri & Thrampoulidis, 2024, Corollary C.1.1 and Proposition C.1) completes the result. $\qquad\square$

# B PROOFS FOR BOUNDS ON DELAYED GENERALIZATION GAP

## B.1 PROOF OF THEOREM 4

**Theorem 5** (Restatement of Theorem 4). *Assume the loss function is 1-self-bounded, Lipschitz and smooth. Let the network's width $m$ be large enough so that $\sqrt{m} \gtrsim \eta \sum_{j=k+1}^{K} \sum_{t=0}^{T-1} \widehat{F}_j(w_j^{(t)})$. Moreover, assume $w_k^\star$ achieving small training loss for task $k$ satisfying $\|w_k^\star - w_k^{(0)}\|^2 \geq \max\{\eta T \widehat{F}_k(w_k^\star), \eta \widehat{F}_k(w_k^{(0)})\}$, and $m \gtrsim \|w_k^\star - w_{k-1}^{(0)}\|^4$. Then,*

$$\mathbb{E}_{\mathcal{D}_k}\left[ F_k(w_K) - \widehat{F}_k(w_K) \right] \leq \frac{\eta \, e^{\frac{\eta}{\sqrt{m}} c_{k,K}}}{n} \sum_{t=0}^{T-1} \mathbb{E}_{\mathcal{D}_k}\left[ \widehat{F}_k(w_k^{(t)}) \right],$$

*where $c_{k,K} = O\left( \sum_{j=k+1}^{K} \sum_{t=0}^{T-1} \widehat{F}_j(w_j^{(t)}) \right)$.*

Recall the continual learning of $K$ tasks for $T$ iterations each i.e., at task $k \in [K]$ :

$$w_t = w_{t-1} - \eta \nabla \widehat{F}_k(w_t) \ \text{ for } (k-1) \cdot T < t \leq k \cdot T$$

Assume $f(\cdot, x)$ to be the sample loss which is $L$-Lipschitz with respect to its first input. Let $\mathcal{D}_k := \{x_1, \cdots, x_n\}$ be the training dataset of task $k$. Denote $w_\ell^{\neg i}$ as the output of the continual learning algorithm after learning task $\ell$ (for some $\ell \leq K$), when $x_i$ is left out of the training samples from $\mathcal{D}_k$. Similarly, we define $w_\ell^{(t), \neg i}$, as the output of continual learning at iteration $t$ of task $\ell$ when $x_i$ is left out.

The generalization gap associated with task $k$, after learning $K$ tasks can be written as,

$$\mathbb{E}_{\mathcal{D}_k}[F_k(w_K) - \widehat{F}_k(w_K)] = \frac{1}{n} \sum_{i=1}^{n} \mathbb{E}_{\mathcal{D}_k, x}\left[ f(w_K, x) - f(w_K, x_i) \right]$$

$$= \frac{1}{n} \sum_{i=1}^{n} \mathbb{E}_{\mathcal{D}_k, x}\left[ f(w_K, x) - f(w_K^{\neg i}, x_i) \right] + \frac{1}{n} \sum_{i=1}^{n} \mathbb{E}_{\mathcal{D}_k}\left[ f(w_K^{\neg i}, x_i) - f(w_K, x_i) \right]$$

$$= \frac{1}{n} \sum_{i=1}^{n} \mathbb{E}_{\mathcal{D}_k, x}\left[ f(w_K, x) - f(w_K^{\neg i}, x) \right] + \frac{1}{n} \sum_{i=1}^{n} \mathbb{E}_{\mathcal{D}_k}\left[ f(w_K^{\neg i}, x_i) - f(w_K, x_i) \right]$$

$$\leq \frac{L}{n} \sum_{i=1}^{n} \mathbb{E}_{\mathcal{D}_k}\left[ \|w_K - w_K^{\neg i}\| \right]. \tag{7}$$

Therefore, the samples from the subsequent distributions do not impact the delayed generalization gap in Task $k$, as we are taking the expectation over only $\mathcal{D}_k$.

$$\|w_K - w_K^{\neg i}\| = \|w_K^{(T-1)} - \eta \nabla \hat{F}_K(w_K^{(T-1)}) - (w_K^{(T-1),\neg i} - \eta \nabla \hat{F}_K(w_K^{(T-1),\neg i}))\|$$

Note that the objectives are the same for both $w_K^{(T-1),\neg i}$ and $w_K^{(T-1)}$. Therefore, by the non-expansive properties of one-hidden-layer neural nets (Taheri & Thrampoulidis, 2024, Lemma B.1):

$$\|w_K - w_K^{\neg i}\| \leq \left(1 + \frac{\eta L R^2}{\sqrt{m}} \max_{w_\alpha \in [w_K^{(T-1)}, w_K^{(T-1),\neg i}]} \widehat{F}'_K(w_\alpha)\right) \|w_K^{(T-1)} - w_K^{(T-1),\neg i}\| \quad (8)$$

where we define:

$$\widehat{F}'(w) := \frac{1}{n} \sum_{i=1}^{n} |f'(w, x_i)|,$$

with $f'$ denoting the derivative of the sample loss. For self-bounded losses assumed in this theorem, we have $|f'(w, x_i)| \leq f(w, x_i)$, therefore $\widehat{F}'_K(w_\alpha) \leq \widehat{F}_K(w_\alpha)$, leading to:

$$\|w_K - w_K^{\neg i}\| \leq \left(1 + \frac{\eta L R^2}{\sqrt{m}} \max_{w_\alpha \in [w_K^{(T-1)}, w_K^{(T-1),\neg i}]} \widehat{F}_K(w_\alpha)\right) \|w_K^{(T-1)} - w_K^{(T-1),\neg i}\|$$

Repeating this step for $T$ steps from $T$ to 1:

$$\|w_K - w_K^{\neg i}\| \leq \prod_{t=0}^{T-1} \left(1 + \frac{\eta L R^2}{\sqrt{m}} \max_{w_{\alpha t} \in [w_K^{(t)}, w_K^{(t),\neg i}]} \widehat{F}_K(w_{\alpha t})\right) \left\|w_K^{(0)} - w_K^{(0),\neg i}\right\|$$

$$= \prod_{t=0}^{T-1} \left(1 + \frac{\eta L R^2}{\sqrt{m}} \max_{w_{\alpha t} \in [w_{K-1}^{(t)}, w_K^{(t),\neg i}]} \widehat{F}_K(w_{\alpha t})\right) \left\|w_K - w_K^{\neg i}\right\|$$

$$\leq \exp\left(\frac{\eta L R^2}{\sqrt{m}} \sum_{t=0}^{T-1} \max_{w_{\alpha t} \in [w_K^{(t)}, w_K^{(t),\neg i}]} \widehat{F}_K(w_{\alpha t})\right) \left\|w_K - w_K^{\neg i}\right\|.$$

where $R$ is the max norm of data and $L$ is the activation function's Lipschitz parameter. We need an inductive argument here to prove that $\|w_t - w_t^{\neg i}\|$ remains bounded for all $t$ as it is used in the max over $w_{\alpha t}$ term.

Repeating this step for $K - k$ tasks, we derive the following,

$$\|w_K - w_K^{\neg i}\| \leq \exp\left(\frac{\eta L R^2}{\sqrt{m}} \sum_{j=k+1}^{K} \sum_{t=0}^{T-1} \max_{w_{\alpha t} \in [w_j^{(t)}, w_j^{(t),\neg i}]} \widehat{F}_j(w_{\alpha t})\right) \left\|w_k - w_k^{\neg i}\right\|. \quad (9)$$

This gives an expression for bounding the generalization gap based on the parameter stability of the $k'$th task, the width and training performance from task $k + 1$ to task $K$. To bound the parameter stability term, note that,

$$\left\|w_k - w_k^{\neg i}\right\| \leq \left\|w_k^{(T-1)} - \eta \nabla \widehat{F}_k(w_k^{(T-1)}) - (w_k^{(T-1),\neg i} - \eta \nabla \widehat{F}_k(w_k^{(T-1),\neg i}))\right\|$$

$$+ \eta \left\|\nabla \widehat{F}_k^i(w_k^{(T-1),\neg i})\right\|$$

Recall the $i$th data point is taken from the $k$th task data distribution. For tasks $j$ where $j < k$, it holds that $w_j = w_j^{\neg i}$. Therefore we can use the result from previous works (Taheri & Thrampoulidis, 2024, Thm B.2) on the stability error of neural networks in the NTK regime to bound $\left\|w_k - w_k^{\neg i}\right\|$.

**Lemma 1.** *If there exists $w_k^\star$ such that $\|w_k^\star - w_k\| \geq \max\left\{\sqrt{\eta T \widehat{F}_k(w_k^\star)}, \sqrt{\eta \widehat{F}_k(w_{k-1})}\right\}$, and*

$m \gtrsim \|w_k^\star - w_k\|^4$, then $\left\|w_k - w_k^{\neg i}\right\| \lesssim \frac{\eta}{n} \sum_{t=0}^{T-1} \widehat{F}_k^i(w_k^{(t)})$, and consequently,

$$\mathbb{E}_{\mathcal{D}_k}\left[\frac{1}{n} \sum_{i=1}^{n} \left\|w_k - w_k^{\neg i}\right\|\right] \lesssim \frac{\eta}{n} \sum_{t=0}^{T-1} \mathbb{E}_{\mathcal{D}_k}\left[\widehat{F}_k(w_k^{(t)})\right].$$

Let us define $c_{k,K} := \max_{i \in [n]} \sum_{j=k+1}^{K} \sum_{t=0}^{T-1} \max_{w_{\alpha t} \in [w_j^{(t)}, w_j^{(t), \neg i}]} \widehat{F}_j(w_{\alpha t})$. Then by this lemma we have,

$$\mathbb{E}_{\mathcal{D}_k}\left[ \frac{1}{n} \sum_{i=1}^{n} \|w_K - w_K^{\neg i}\| \right] \leq \frac{\eta \, e^{\frac{\eta}{\sqrt{m}} c_{k,K}}}{n} \sum_{t=0}^{T-1} \mathbb{E}_{\mathcal{D}_k}\left[ \widehat{F}_k(w_k^{(t)}) \right].$$

### B.1.1   Bounding $c_{k,K}$

In order to bound $c_{k,K}$, we use the following result on the quasi-convexity properties of the two-layer neural net objective by (Taheri & Thrampoulidis, 2024, Prop. 5.1.).

**Lemma 2.** *Suppose $\widehat{F} : \mathbb{R}^{d'} \to \mathbb{R}$ satisfies the self-bounded weak convexity property with parameter $\kappa$. Let $w_1, w_2 \in \mathbb{R}^{d'}$ be two arbitrary points with distance $\|w_1 - w_2\| \leq D < \sqrt{2/\kappa}$. Set $\tau := \left(1 - \kappa D^2/2\right)^{-1}$. Then,*

$$\max_{v \in [w_1, w_2]} \widehat{F}(v) \leq \tau \cdot \max\left\{ \widehat{F}(w_1), \widehat{F}(w_2) \right\}.$$

For self-bounded losses $\kappa = \frac{1}{\sqrt{m}}$, therefore if $w, w'$ are such that $\|w - w'\| \leq D \leq m^{1/4}$, then

$$\max_{v \in [w, w']} \widehat{F}(v) \leq \frac{1}{1 - \frac{D^2}{\sqrt{m}}} \cdot \max\left\{ \widehat{F}(w), \widehat{F}(w') \right\}.$$

Recall,

$$\|w_K - w_K^{\neg i}\| \leq \exp\left( \frac{\eta}{\sqrt{m}} \sum_{j=k+1}^{K} \sum_{t=0}^{T-1} \max_{w_{\alpha t} \in [w_j^{(t)}, w_j^{(t), \neg i}]} \widehat{F}_j(w_{\alpha t}) \right) \|w_k - w_k^{\neg i}\|.$$

Assume

$$\sqrt{m} \geq 8 \max\left\{ \eta \sum_{j=k+1}^{K} \sum_{t=0}^{T-1} (\widehat{F}_j(w_j^{(t)}) + \widehat{F}_j(w_j^{(t), \neg i})), \|w_k - w_k^{\neg i}\|^2 \right\}. \tag{10}$$

Then, by induction $\|w_j^{(t)} - w_j^{(t), \neg i}\| \leq 2\|w_k - w_k^{\neg i}\|$ for all $t \in [T], j \in [k, K]$. To see this:

$$\|w_j^{(t)} - w_j^{(t), \neg i}\|$$

$$\leq \exp\left( \frac{\eta}{\sqrt{m}} \sum_{j'=k+1}^{j-1} \sum_{\tau=0}^{T-1} \max_{w_{\alpha \tau} \in [w_{j'}^{(\tau)}, w_{j'}^{(\tau), \neg i}]} \widehat{F}_{j'}(w_{\alpha \tau}) + \sum_{\tau=0}^{t-1} \max_{w_{\alpha \tau} \in [w_j^{(\tau)}, w_j^{(\tau), \neg i}]} \widehat{F}_j(w_{\alpha \tau}) \right)$$

$$\times \|w_k - w_k^{\neg i}\|$$

By induction's assumption $\sqrt{m} \geq 2\|w_{j'}^{(\tau)} - w_{j'}^{(\tau), \neg i}\|^2$. Therefore we can invoke Lemma 2 for all the $\max \widehat{F}_{j'}$ to find that,

$$\|w_j^{(t)} - w_j^{(t), \neg i}\| \leq \exp(1/4) \cdot \|w_k - w_k^{\neg i}\| \leq 2 \|w_k - w_k^{\neg i}\|.$$

Which proves the induction. Overall, we could bound $c_{K,k}$ based on the training objective. assuming $\widehat{F}_j(w_j^{(t)})$ and $\widehat{F}_j(w_j^{(t), \neg i})$ are of the same order(needs proof), then we find

$$c_{K,k} \leq 2 \sum_{j=k+1}^{K} \sum_{t=0}^{T-1} (\widehat{F}_j(w_j^{(t)}) + \widehat{F}_j(w_j^{(t), \neg i})) = O\left( \sum_{j=k+1}^{K} \sum_{t=0}^{T-1} \widehat{F}_j(w_j^{(t)}) \right)$$

To simplify the statement of the lemma, we can assume $\widehat{F}_j(w_j^{(t)})$ and $\widehat{F}_j(w_j^{(t), \neg i})$ are of the same order as reducing the sample-size by 1 sample does not affect the training bounds.

**Lemma 3.** *Let the assumptions of Lemma 1 hold. Assume*

$$\sqrt{m} \gtrsim \eta \sum_{j=k+1}^{K} \sum_{t=0}^{T-1} (\widehat{F}_j(w_j^{(t)}) + \widehat{F}_j(w_j^{(t),\neg i})) \asymp \eta \sum_{j=k+1}^{K} \sum_{t=0}^{T-1} \widehat{F}_j(w_j^{(t)}).$$

*Then,*

$$\mathbb{E}_{\mathcal{D}_k} \left[ \frac{1}{n} \sum_{i=1}^{n} \| w_K - w_K^{\neg i} \| \right] \leq \frac{\eta}{n} \mathbb{E}_{\mathcal{D}_k} \left[ e^{\frac{\eta}{\sqrt{m}} c_{k,K}} \sum_{t=0}^{T-1} \widehat{F}_k(w_k^{(t)}) \right]$$

*where $c_{k,K} = O\left( \sum_{j=k+1}^{K} \sum_{t=0}^{T-1} \widehat{F}_j(w_j^{(t)}) \right)$.*

*Proof.* The proof essentially follows by the last two lemmas and noting that $\| w_k - w_k^{\neg i} \| \leq \| w_k - w_{k-1} \| + \| w_k^{\neg i} - w_{k-1} \| = O(\| w_k^{\star} - w_{k-1} \|)$ by Lemma 1. Therefore the condition we had in Eq. 10 on $\sqrt{m} \geq \| w_k - w_k^{\neg i} \|^2$ is absorbed in the condition from Lemma 1.

$\qquad\square$

This completes the proof of Theorem 4.

### B.2 Proof of Theorem 3

**Theorem 6** (Restatement of Theorem 3)**.** *Assume the loss function is 1-Lipschitz and 1-smooth. Then, the expected delayed generalization gap satisfies,*

$$\mathcal{F}_{k,K}^{\text{gen}} := \mathbb{E}_{\mathcal{D}_k} \left[ F_k(w_K) - \widehat{F}_k(w_K) \right] \lesssim \frac{\eta T \, e^{\frac{\eta T(K-k+1)}{\sqrt{m}}}}{n}.$$

*Proof.* The proof of Theorem 3 essentially follows from Theorem 4. We outline the distinct steps. Note that since the objective is 1-Lipschitz, it holds $\widehat{F}'(w) \leq 1$ for any $w$.. Therefore Eq. 8 from the proof of Theorem 3 changes into

$$\| w_K - w_K^{\neg i} \| \leq \left( 1 + \frac{\eta L R^2}{\sqrt{m}} \right) \left\| w_K^{(T-1)} - w_K^{(T-1),\neg i} \right\|.$$

As a result, by unrolling the iterates and noting that $R \leq 1$:

$$\left\| w_K - w_K^{\neg i} \right\| \leq \exp \left( \frac{\eta L (K-k) T}{\sqrt{m}} \right) \left\| w_k - w_k^{\neg i} \right\|. \tag{11}$$

Moreover, again using the Lipschitz loss function properties:

$$
\begin{aligned}
\left\| w_k - w_k^{\neg i} \right\| &\leq \left\| w_k^{(T-1)} - \eta \nabla \widehat{F}_k(w_k^{(T-1)}) - (w_k^{(T-1),\neg i} - \eta \nabla \widehat{F}_k(w_k^{(T-1),\neg i})) \right\| \\
&\quad + \eta \left\| \nabla \widehat{F}_k^i(w_k^{(T-1),\neg i}) \right\| \\
&\leq \left\| w_k^{(T-1)} - \eta \nabla \widehat{F}_k(w_k^{(T-1)}) - (w_k^{(T-1),\neg i} - \eta \nabla \widehat{F}_k(w_k^{(T-1),\neg i})) \right\| + \frac{\eta L}{n} \\
&\leq \exp(\frac{\eta L}{\sqrt{m}}) \| w_k^{(T-1)} - w_k^{(T-1),\neg i} \| + \frac{\eta L}{n} \\
&\leq \exp(\frac{2\eta L}{\sqrt{m}}) \| w_k^{(T-2)} - w_k^{(T-2),\neg i} \| + (1 + \exp(\frac{\eta L}{\sqrt{m}})) \frac{\eta L}{n} \\
&\leq (\sum_{t=0}^{T-1} \exp(\frac{\eta L t}{\sqrt{m}})) \frac{\eta L}{n} \\
&\leq \exp(\frac{\eta L T}{\sqrt{m}}) \frac{\eta L T}{n}.
\end{aligned}
$$

where the last step is derived by repeating the procedure over all $T$ iterations.

Inserting this in Eq. 11, taking the expectation over $\mathcal{D}_k$, using Eq. 7 and noting that $L$ (the objective's Lipschitz parameter) is constant for our setup, conclude the proof of the theorem. □

## C    Bounding Train-time Loss and Forgetting for XOR cluster data

In this section, we prove Theorems 1-2. Below, is a restatement of these theorems.

**Theorem 7** (Restatement of Theorems 1-2). *Consider the $d$-dimensional XOR cluster dataset with $K$ tasks and assume gradient descent with $\eta T = \Theta(d^2)$ iterations and $n = \widetilde{\Theta}(d^2 K)$ samples for each subsequent task trained by a neural net with $m = \widetilde{\Omega}(d^8 K^4)$ hidden neurons. Then, with high probability, the train-time forgetting and per-task train-time time error is $\mathcal{F}_{k,K}^{\mathrm{tr}} = o_d(1)$. In particular, for the train-time forgetting with probability $1 - \delta$, we have:*

$$|\mathcal{F}_{k,K}^{\mathrm{tr}}| := |\widehat{F}_k(w_K) - \widehat{F}_k(w_k)| = \widetilde{O}\left(\eta T \frac{\sqrt{K-k}}{d\sqrt{n}} + \eta T \frac{\sqrt{K-k}}{d^2 \operatorname{poly}\log(d)} + \eta^2 T^2 \frac{K^2}{\sqrt{m}}\right),$$

*where $\widetilde{O}(\cdot)$ hides logarithmic factors in $n$ and $\delta$.*

The proof strategy is as follows. First, we consider the $m \to \infty$ and derive the weights for arbitraay number of GD steps for each task. We then show that for sufficiently large $T$ and sufficiently large $n$, and by computing the network output via concentration bounds based on $n$ for the considered XOR cluster dataset, the train-loss and forgetting are approximately zero. We then compute the error due to finite-width, showing that under sufficiently small $T$, and sufficiently large $m$, the derivations of the infinite-width regime are approximately correct. This leads to the desired quantities and train-time forgetting bounds based $n, T$ and $m$ as stated in the theorem. We start by considering the infinite width regime.

### C.1    Training error for an infinitely wide network

First, we consider the $m \to \infty$ regime and characterize the distribution of the final weights after $T$ and $2T$ iterations in this regime. We then discuss the general formula for arbitrary number of tasks. Recall, we considered the hinge-loss for training-time analysis. However, as mentioned in the main body of the paper and as it will become clear in the following analysis, we can simplify the arguments by noting that throughout the optimization process for all $K$ tasks, only the linear part of the loss is used. Thus we can assume the loss function as $f(u) = 1 - u$ without loss of generality.

Let us simplify the notation by droping the task index from weights and instead denoting the vector entering the $i$th neuron by $w^i \in \mathbb{R}^d$. Note that by Taylor expansion around the Gaussian initialization $w_9$, we have,

$$\Phi(w, x) = \Phi(w_0, x) + \frac{1}{\sqrt{m}} \sum_{i=1}^{m} a_i \phi'(\langle w_0^i, x \rangle) \langle x, w^i - w_0^i \rangle + O(\frac{\|w - w_0\|}{\sqrt{m}}).$$

For $w$ close to $w_0$, and for large enough $m$ we can use a linearized neural network model. In particular, in the $m \to \infty$ regime, the updates of the continual learning algorithm are the following for sufficiently small $T$:

$$w_1^i = w_0^i + \eta \frac{1}{\sqrt{m}} \frac{1}{n} \sum_{j=1}^{n} a_i \phi'(\langle w_0^i, x_j^1 \rangle) x_j^1 y_j^1$$

$$w_T^i = w_0^i + \frac{\eta T}{\sqrt{m}} \frac{1}{n} \sum_{j=1}^{n} a_i \phi'(\langle w_0^i, x_j^1 \rangle) x_j^1 y_j^1$$

$$w_{2T}^i = w_0^i + \frac{\eta T}{\sqrt{m}} \frac{1}{n} \sum_{j=1}^{n} a_i \phi'(\langle w_0^i, x_j^1 \rangle) x_j^1 y_j^1 + \frac{\eta T}{\sqrt{m}} \frac{1}{n} \sum_{j=1}^{n} a_i \phi'(\langle w_0^i, x_j^2 \rangle) x_j^2 y_j^2$$

We consider $x_j^k, y_j^k$ for any $j \in [n]$ and $k \in [K]$ as fixed training points used for training task $k$. We consider randomness only with respect to the initialization $w_0^i$ and characterize the distribution of weights in the infinite width regime. As $m \to \infty$ given the IID initialization for $w_0^i$ and the quadratic activation, we deduce the following convergence in distribution,

$$w_T^i = w_0^i + \frac{\eta T}{\sqrt{m}} \frac{1}{n} \sum_{j=1}^n a_i \phi'(\langle w_0^i, x_j \rangle) x_j^1 y_j^1 \to \omega z + \frac{\eta t}{\sqrt{m}} \frac{1}{n} \sum_{j=1}^n z^\top x_j^1 \, x_j^1 y_j^1 \tag{12}$$

where $\omega \in \{\pm 1\}, z \in \mathbb{R}^d$ are Rademacher random variable and standard Gaussian random vector, respectively, and they represent first layer and second layer initialization.

Let us briefly consider the matrix formulation,

$$R = \frac{1}{n} \sum_{j=1}^n y_j^1 x_j^1 z^\top x_j^1 =: Az$$

then $R \sim \mathcal{N}(0, A^2)$ as $Cov(R) = \mathbb{E}[Azz^\top A^\top] = A\mathbb{E}[zz^\top]A^\top = AA^\top = A^2$. In the infinite $n$ asymptotic, $A \to \mathbb{E}[y_j x_j x_j^\top] = \frac{1}{2}\mathbb{E}[xx^\top | y = 1] - \frac{1}{2}\mathbb{E}[xx^\top | y = -1] = \frac{1}{2}(\mu_+^1 \mu_+^{1\top} - \mu_-^1 \mu_-^{1\top})$, indicating that the GD updates learn the true vectors in the $n \to \infty$ regime.

A similar argument leads to the following update rule for the second task:

$$w_{2T}^i \sim z + \frac{\eta T}{\sqrt{m}} A_1 \omega z + \frac{\eta T}{\sqrt{m}} A_2 \omega z, \quad A_1 := \frac{1}{n} \sum_{j=1}^n y_j^1 x_j^1 x_j^{1\top}, \quad A_2 := \frac{1}{n} \sum_{j=1}^n y_j^2 x_j^2 x_j^{2\top}$$

where again $z \sim \mathcal{N}(0, I_d)$ and $\omega$ is a Rademacher r.v. for representing the binary second layer weights $a_i$.

Similarly, we find that after $K$ tasks with $T$ iterations for each task, the weight $w_{KT}^i$ takes the following form:

$$w_{KT}^i \sim z + \frac{\eta T}{\sqrt{m}} \sum_{j=1}^K A_j \omega z, \quad A_j := \frac{1}{n} \sum_{v=1}^n y_v^j x_v^j x_v^{j\top}$$

Recalling the expression for the neural network output, we can characterize the output of the network with this random variable in the infinitely wide regime:

$$\Phi(w_{KT}, x) = \frac{1}{\sqrt{m}} \sum_{i=1}^m a_i (\langle w_{KT}^i, x \rangle)^2 \sim \frac{1}{\sqrt{m}} \sum_{i=1}^m \omega_i \left( \left\langle z_i + \frac{\eta T}{\sqrt{m}} \sum_{j=1}^K A_j \omega_i z_i, x \right\rangle \right)^2$$

$$= \frac{\eta T}{m} \sum_{i=1}^m \langle z_i, x \rangle \left\langle (\sum_{j=1}^K A_j) z_i, x \right\rangle + \frac{1}{\sqrt{m}} \sum_{i=1}^m \omega_i (z_i^\top x)^2$$

$$+ \frac{\eta^2 T^2}{m\sqrt{m}} \sum_{i=1}^m \omega_i \left( \left\langle z_i + \frac{\eta T}{\sqrt{m}} \sum_{j=1}^K A_j \omega_i z_i, x \right\rangle \right)^2$$

when $m \to \infty$:

$$\longrightarrow \eta T \, \mathbb{E}_z \left[ \langle z, x \rangle \left\langle (\sum_{j=1}^K A_j) z, x \right\rangle \right] + N + 0$$

$$= \eta T \, x^\top (\sum_{j=1}^K A_j) x + N \tag{13}$$

where the last step is by the law of large number and $N$ denotes the asymptotic distribution of the second term. The last term vanishes by the law of large numbers. We derive the training loss by calculating the above for $x$ coming from the training distribution.

We discuss the role of each term in Eq. 13. First, considering the first term above, the training loss for task $K$ w.r.t the first training sample is the following,

$$\frac{\eta T}{n} {x_1}^{K\top} \sum_{k=1}^{K} \sum_{i=1}^{n} y_i^k x_i^k {x_i^k}^\top x_1^K \tag{14}$$

We split the summation into the relevant task $k = K$ and other tasks when $k \neq K$.

**Case I:** $k = K$**.** Let us drop $K$ in Eq. 14. we have

$$\frac{1}{n} x_1^\top \sum_{i=1}^{n} y_i x_i x_i^\top x_1 = \frac{1}{n} \sum_{i=1}^{n} y_i (x_i^\top x_1)^2 = \frac{1}{n} \left( y_1 \|x_1\|^4 + \sum_{i=2}^{n} y_i (x_i^\top x_1)^2 \right).$$

Recall our data model:

$$x \sim \mathcal{N}(\pm \mu_+^K, \sigma^2 I_d) \quad \text{if } y = +1, \qquad x \sim \mathcal{N}(\pm \mu_-^K, \sigma^2 I_d) \quad \text{if } y = -1,$$

with the following assumptions:

$$\mu_+^K \perp \mu_-^K, \quad \|\mu_+^K\| = \|\mu_+^K\| = \frac{1}{\sqrt{d}}, \quad \sigma = O\left( \frac{1}{\sqrt{d} \operatorname{poly} \log(d)} \right).$$

Let

$$U := \frac{1}{n} \sum_{i=1}^{n} y_i (x_i^\top x_1)^2.$$

Fix $x_1, y_1$. For any $i \neq 1$, we write

$$x_1 = \mu_{y_1}^K + \varepsilon_1, \qquad x_i = \mu_{y_i}^K + \varepsilon_i,$$

with $\varepsilon_1, \varepsilon_i \sim \mathcal{N}(0, \sigma^2 I_d)$, independent. Then:

$$x_i^\top x_1 = {\mu_{y_i}^K}^\top \mu_{y_1}^K + {\mu_{y_i}^K}^\top \varepsilon_1 + {\mu_{y_1}^K}^\top \varepsilon_i + \varepsilon_i^\top \varepsilon_1.$$

Note that $({\mu_{y_i}^K}^\top \mu_{y_1}^K)^2 = \frac{1}{d^2}$ if $y_i = y_1$ and 0 otherwise.

Hence,

$$\mathbb{E}\left[ y_i (x_i^\top x_1)^2 \mid y_i \right] = \begin{cases} \mathbb{E}[y_1(\pm\frac{1}{d} \pm {\mu_{y_1}^K}^\top \varepsilon_1 + {\mu_{y_1}^K}^\top \varepsilon_i + \varepsilon_i^\top \varepsilon_1)^2] & \text{if } y_i = y_1, \\[2mm] \mathbb{E}[-y_1(\pm{\mu_{-y_1}^K}^\top \varepsilon_1 + {\mu_{y_1}^K}^\top \varepsilon_i + \varepsilon_i^\top \varepsilon_1)^2] & \text{if } y_i \neq y_1. \end{cases}$$

Assuming a balanced distribution, i.e., $\Pr[y_i = y_1] = \Pr[y_i \neq y_1] = \frac{1}{2}$, we get:

$$\mathbb{E}\left[ y_i (x_i^\top x_1)^2 \right] = \frac{y_1}{2d^2} + O\left( \frac{1}{d^2 \cdot \operatorname{poly} \log(d)} \right)$$

where in the above, we used ${\mu_{y_1}^K}^\top \varepsilon_1 = O(\frac{1}{d \cdot \operatorname{poly} \log(d)})$ w.h.p. over the randomness in $\epsilon_1$.

Thus, the overall expectation is the following:

$$\mathbb{E}[U] = \frac{y_1}{2d^2} + O\left( \frac{1}{d^2 \cdot \operatorname{poly} \log(d)} \right)$$

which aligns with the true label $y_1$.

To compute the finite sample guarantees, note that each summand

$$Z_i = y_i(x_i^\top x_1)^2$$

is sub-exponential with scale parameter $O(1/d)$ as $(\epsilon_i^\top \epsilon_1)^2$ has standard deviation $\frac{1}{d\operatorname{poly}\log(d)}$ uniformly for all $i > 1$. By Bernstein's inequality, for any $\delta \in (0,1)$, with probability at least $1 - \delta$ over the randomness in $\{x_i, y_i\}_{i \in [n]}$,

$$|U - \mathbb{E}[U]| \leq C\left(\frac{1/d}{\sqrt{n}}\sqrt{\log(1/\delta)} + \frac{1/d}{n}\log(1/\delta)\right) = O\left(\frac{1}{d\sqrt{n}}\sqrt{\log(1/\delta)}\right),$$

for some absolute constant $C > 0$.

Putting together, with probability at least $1 - \delta$

$$U = \frac{1}{n}\sum_{i=1}^{n} y_i(x_i^\top x_1)^2 = \frac{y_1}{2d^2} \pm O\left(\frac{1}{d^2 \cdot \operatorname{poly}\log(d)} + \frac{1}{d\sqrt{n}}\sqrt{\log(1/\delta)}\right).$$

In particular, if $n \gg d^2 \log(1/\delta)$, then the error term is much smaller than the signal $\frac{1}{2d^2}$, and therefore $\operatorname{sign}(T) = y_1$. with a union bound over all training points which introduces an additional factor $\log(n)$ in the above bound, we find that the train error is exactly zero.

**Case II: $k \neq K$.** Now we evaluate the other terms in the summation in Eq. 14

$$x^\top A_j x = \frac{1}{n}\sum_{i=1}^{n} y_i^j(x^\top x_i^j)^2$$

for some $j \neq K$. drop 1 and note that

$$x_i \sim \mathcal{N}(\pm\mu_+^j, \ \sigma^2 I_d) \quad \text{if } y_i = +1, \quad x_i \sim \mathcal{N}(\pm\mu_-^j, \ \sigma^2 I_d) \quad \text{if } y_i = -1,$$
$$x \sim \mathcal{N}(\pm\mu_+^K, \ \sigma^2 I_d) \quad \text{if } y = +1, \quad x \sim \mathcal{N}(\pm\mu_-^K, \ \sigma^2 I_d) \quad \text{if } y = -1,$$

where $\mu_+^j, \mu_-^j, \mu_+^K, \mu_-^K$ are mutually orthogonal, $\|\mu_+^j\| = \|\mu_-^j\| = \|\mu_+^K\| = \|\mu_-^K\| = \frac{1}{\sqrt{d}}$, and $\sigma = O\left(\frac{1}{\sqrt{d}\operatorname{poly}\log(d)}\right)$.

Let

$$U' = \frac{1}{n}\sum_{i=1}^{n} y_i(x_i^\top x)^2.$$

let $x = \mu + \epsilon$, we have

$$\mathbb{E}\left[y_i(x_i^\top x)^2 \mid y_i\right] = \begin{cases} \mathbb{E}[(\pm\mu_{y_i}^{K\top}\varepsilon + \mu^\top\varepsilon_i + \varepsilon_i^\top\varepsilon)^2] & \text{if } y_i = 1, \\ \mathbb{E}[-(\pm\mu_{-y_i}^{K\top}\varepsilon + \mu^\top\varepsilon_i + \varepsilon_i^\top\varepsilon)^2] & \text{if } y_i = -1. \end{cases}$$

Hence,

$$\mathbb{E}[U'] = O(\frac{1}{d^2\operatorname{poly}\log(d)}).$$

Define

$$Z_i = y_i(x_i^\top x)^2.$$

By expanding $x_i = \mu_{y_i}^j + \varepsilon_i$, $x = \mu_y^K + \varepsilon'$, and using $\sigma = O(1/\sqrt{d})$, one can verify that

$$\operatorname{Var}(x_i^\top x) = O\left(\frac{1}{d}\right),$$

and that $(x_i^\top x)^2$ is sub-exponential with scale parameter $O(1/d)$. Thus each $Z_i$ is sub-exponential with parameter $O(1/d)$.

By Bernstein's inequality for i.i.d. sub-exponential random variables, for any $\delta \in (0,1)$, with probability at least $1 - \delta$,

$$|U'| = \left|\frac{1}{n}\sum_{i=1}^{n}(Z_i - \mathbb{E}[Z_i])\right| \leq C\left(\frac{1/d}{\sqrt{n}}\sqrt{\log(1/\delta)} + \frac{1/d}{n}\log(1/\delta)\right) = O\left(\frac{1}{d\sqrt{n}}\sqrt{\log(1/\delta)}\right),$$

for some absolute constant $C$.

**Combining the two cases.** Together with the two results above we find for any training data point $(x, y)$ from task $K$:

$$x^\top (\sum_{j=1}^{K} A_j)x = \frac{y}{2d^2} \pm O\left(\frac{\sqrt{K}}{d^2 \operatorname{poly} \log(d)} + \frac{\sqrt{K}}{d\sqrt{n}}\sqrt{\log(1/\delta)}\right)$$

This concludes the calculations of the first term in Eq. 13.

Now let us consider the noise term (denoted by N) in Eq. 13:

$$N = \frac{1}{\sqrt{m}} \sum_{i=1}^{m} \omega_i (z_i^\top x)^2.$$

note that $\omega_i(z_i^\top x)^2$ has variance $O(\frac{1}{\operatorname{poly} \log(d)})$, therefore by CLT

$$\frac{1}{\sqrt{m}} \sum_{i=1}^{m} \omega_i (z_i^\top x)^2 \to \mathcal{N}(0, \frac{1}{\operatorname{poly} \log(d)}).$$

Overall, in the infinite width limit, for some $x, y$ from the $K$th task's empirical distribution

$$\Phi(w_{KT}, x) = \eta T \, x^\top (\sum_{j=1}^{K} A_j)x + \mathcal{N}(0, 1)$$

$$= \eta T \left(\frac{y_k}{d^2} \pm O(\frac{\sqrt{K}}{d^2 \operatorname{poly} \log(d)} + \frac{\sqrt{K}}{d\sqrt{n}}\sqrt{\log(1/\delta)})\right) + O\left(\frac{\sqrt{\log(1/\delta)}}{\operatorname{poly} \log(d)}\right).$$

In particular, if $n = \Omega(d^2 K \log(1/\delta))$, then the error term is smaller than the signal $\frac{y_k}{d^2}$, and if $\eta T = \Theta(d^2)$ then the output aligns with $y$. With a union bound over all training points (which introduces an additional factor $\log(n)$ in the above bound), we find that the train error (%) is exactly zero for all $k \in [n]$, leading to the zero train error.

## C.2 CHARACTERIZING FORGETTING FOR INFINITELY WIDE NETS

We can directly compute $\widehat{F}_k(w_{KT})$ by computing $\Phi(w_{KT}, x_1^k)$ where $x_1^k$ is a sample (first sample w.l.o.g) from the training data for task $k$ where $k < K$. Recall,

$$w_{KT}^i \sim z + \frac{\eta T}{\sqrt{m}} \sum_{j=1}^{K} A_j \omega z, \quad A_j := \frac{1}{n} \sum_{v=1}^{n} y_v^j x_v^j {x_v^j}^\top$$

note that the above is symmetric with respect to the task index therefore $\lim_{m \to \infty} \Phi(w_{KT}, x_1^k) = \lim_{m \to \infty} \Phi(w_{KT}, x_1^K)$ in distribution. and we have in the $m \to \infty$ limit for $x_k := x_k^1$:

$$\Phi(w_{KT}, x_k) = \eta T \, x_k^\top (\sum_{j} A_j)x_k + \mathcal{N}(0, \frac{1}{\operatorname{poly} \log(d)}) \tag{15}$$

$$= \eta T \left(\frac{y_k}{d^2} \pm O\left(\frac{\sqrt{K}}{d^2 \operatorname{poly} \log(d)} + \frac{\sqrt{K}}{d\sqrt{n}}\sqrt{\log(1/\delta)}\right)\right) + O\left(\frac{\sqrt{\log(1/\delta)}}{\operatorname{poly} \log(d)}\right).$$

Therefore, again if $n = \Omega(d^2 K \log(1/\delta))$, then the error term is smaller than the signal $\frac{y_k}{d^2}$, and if $\eta T = \Theta(d^2)$ then the output aligns with $y_k$. With a union bound over all training points $k \in [n]$, we find that the training error is exactly zero for all tasks.

Now to characterize forgetting, recall it is defined as

$$|\widehat{F}_k(w_K) - \widehat{F}_k(w_k)| = |\sum_{x_k} \eta T \, x_k^\top (\sum_{j=k+1}^{K} A_j)x_k| \tag{16}$$

$$= \eta T \cdot O\left(\frac{\sqrt{K-k}}{d^2 \operatorname{poly} \log(d)} + \frac{\sqrt{K-k}}{d\sqrt{n}}\sqrt{\log(1/\delta)}\right).$$

where the calculations are the same as before except that the impact of initialization noise is present in both $\widehat{F}_k(w_K), \widehat{F}_k(w_k)$ and thus it is canceled.

In the above expression, if $n = \widetilde{\Omega}(d^2(K-k))$ and $\eta T \asymp d^2$, the increase in forgetting is $o_d(1)$.

## C.3 FINITE-WIDTH ERROR

The calculations above hold for the infinitely-wide network. In this section, we derive the error due to finite width. Recall,

$$\Phi(w, x) = \Phi(w_0, x) + \frac{1}{\sqrt{m}} \sum_{i=1}^{m} a_i \phi'(\langle w_0^i, x \rangle) \langle x, w^i - w_0^i \rangle + O(\frac{\|w - w_0\|^2}{\sqrt{m}})$$

for the infinite width limit we had,

$$\bar{w}_1^i = w_0^i + \eta \frac{1}{\sqrt{m}} \frac{1}{n} \sum_{j=1}^{n} a_i \phi'(\langle w_0^i, x_j^1 \rangle) x_j^1 y_j^1$$

$$\bar{w}_t^i = w_0^i + \frac{\eta t}{\sqrt{m}} \frac{1}{n} \sum_{j=1}^{n} a_i \phi'(\langle w_0^i, x_j^1 \rangle) x_j^1 y_j^1$$

$$\bar{w}_{2t}^i = w_0^i + \frac{\eta t}{\sqrt{m}} \frac{1}{n} \sum_{j=1}^{n} a_i \phi'(\langle w_0^i, x_j^1 \rangle) x_j^1 y_j^1 + \frac{\eta t}{\sqrt{m}} \frac{1}{n} \sum_{j=1}^{n} a_i \phi'(\langle w_0^i, x_j^2 \rangle) x_j^2 y_j^2,$$

and similarly, all tasks' updates were derived. Let $\Phi(\cdot, \cdot)$ be the infinite-width and $\Phi_m(\cdot, \cdot)$ be the finite-width formulations of the network output. Then, we are interested in bounding $|\Phi(\bar{w}_t, x) - \Phi_m(w_t, x)|$ which can be written as:

$$|\Phi(\bar{w}_t, x) - \Phi_m(w_t, x)| \leq |\Phi(z, x) - \Phi_m(w_0, x)|$$

$$+ \Big| \frac{1}{\sqrt{m}} \sum_{i=1}^{m} a_i \langle w_0^i, x \rangle \langle x, w_t^i - w_0^i \rangle - \eta t \mathbb{E}_z[\langle z, x \rangle \langle x, A_1 z \rangle] \Big|$$

$$+ O(\frac{\|w_t - w_0\|^2}{\sqrt{m}})$$

$$\leq O(\frac{1}{\sqrt{m}} + \frac{\|w_t - w_0\|^2}{\sqrt{m}})$$

$$+ \Big| \frac{1}{\sqrt{m}} \sum_{i=1}^{m} a_i \langle w_0^i, x \rangle \langle x, w_t^i - w_0^i \rangle - \eta t \mathbb{E}_z[\langle z, x \rangle \langle x, A_1 z \rangle] \Big|$$

where we used the fact that by LLN: $\frac{1}{\sqrt{m}} \sum_{i=1}^{m} a_i \langle w_0^i, x \rangle \langle x, w_t^i - w_0^i \rangle \to \eta t \mathbb{E}_z[\langle z, x \rangle \langle x, A_1 z \rangle]$.

Note that $w_t^i - w_0^i = \frac{\eta}{\sqrt{mn}} \sum_{\tau=0}^{t-1} \sum_{j=1}^{n} a_i \langle w_\tau^i, x_j \rangle x_j y_j$, therefore when $m \to \infty$:

$$\frac{1}{\sqrt{m}} \sum_{i=1}^{m} a_i \langle w_0^i, x \rangle \langle x, w_t^i - w_0^i \rangle = \frac{\eta}{n} \sum_{\tau=0}^{t-1} \sum_{j=1}^{n} \langle x, x_j y_j \rangle \frac{1}{m} \sum_{i=1}^{m} \langle w_0^i, x \rangle \langle w_\tau^i, x_j \rangle$$

$$\to \frac{\eta}{n} \sum_{\tau=0}^{t-1} \sum_{j=1}^{n} \langle x, x_j y_j \rangle \mathbb{E}[\langle w_0^i, x \rangle \langle w_\tau^i, x_j \rangle]$$

$\langle w_0^i . x \rangle$ is Gaussian with variance $\|x\|^2$ and $\langle w_\tau^i, x_j \rangle$ is bounded by $D_\tau^i \|x_j\|$ where $D_\tau^i := \|w_\tau^i - w_0^i\|$, therefore $\langle w_0^i, x \rangle \langle w_\tau^i, x_j \rangle$ is bounded by $\|x\| \|x_j\| D_\tau^i = O(D_\tau^i)$. and by Hoeffding's concentration inequality:

$$\Big| \frac{1}{m} \sum_{i=1}^{m} \langle w_0^i, x \rangle \langle w_\tau^i, x_j \rangle - \mathbb{E}[\langle w_0^i, x \rangle \langle w_\tau^i, x_j \rangle] \Big| = O(\frac{D_\tau^i}{\sqrt{m}})$$

and hence w.h.p,

$$\left| \frac{\eta}{n} \sum_{\tau=0}^{t-1} \sum_{j=1}^{n} \langle x, x_j y_j \rangle \frac{1}{m} \sum_{i=1}^{m} \langle w_0^i, x \rangle \langle w_\tau^i, x_j \rangle - \frac{\eta}{n} \sum_{\tau=0}^{t-1} \sum_{j=1}^{n} \langle x, x_j y_j \rangle \mathbb{E}[\langle w_0^i, x \rangle \langle w_\tau^i, x_j \rangle] \right|$$

$$= O\left( \frac{\eta}{\sqrt{m}} \sum_\tau \max_i D_\tau^i \right)$$

$$= \widetilde{O}\left( \frac{\eta}{\sqrt{m}} \sum_\tau D_\tau^1 \right) = \widetilde{O}\left( \frac{\eta t D_t^1}{\sqrt{m}} \right)$$

where we used again $x^\top x_j \lesssim 1$, the fact that due to symmetry we expect $D_\tau^i$ to be of the same order for different $i$ s and also $D_\tau^i < D_t^i$ for all $\tau \leq t$. Putting these back to the inequality in the last page for the finite-width error of the network's output:

$$|\Phi(\bar{w}_t, x) - \Phi_m(w_t, x)| = \widetilde{O}\left( \frac{1}{\sqrt{m}} + \frac{\|w_t - w_0\|^2}{\sqrt{m}} + \frac{\eta t \|w_t^1 - w_0^1\|}{\sqrt{m}} \right).$$

similarly

$$|\Phi(\bar{w}_{KT}, x) - \Phi_m(w_{KT}, x)| = \widetilde{O}\left( \frac{1}{\sqrt{m}} + \frac{\|w_{KT} - w_0\|^2}{\sqrt{m}} + \frac{\eta KT \|w_{KT}^1 - w_0^1\|}{\sqrt{m}} \right). \quad (17)$$

### C.3.1 BOUNDING THE WEIGHTS DISTANCE FROM INITIALIZATION

In order to complete the proof, we need to bound the distance from initialization i.e., $\|w_t - w_0\|$ and $\|w_t^i - w_0^i\|$ for every $i$ and $t$. We do this by an iterative argument as follows. Note that for the XOR cluster dataset $\|x\| = \Theta_d(1)$. Then, by recalling the updates of GD, we find that,

$$\|w_1^i - w_0^i\| \leq \frac{\eta}{\sqrt{m}n} \sum_{i=1}^{n} |\langle w_0^i, x_j^1 \rangle| \|x_j^1\| \lesssim \frac{\eta}{\sqrt{m}}$$

$$\|w_2^i - w_0^i\| \leq \frac{\eta}{\sqrt{m}n} \sum_{i=1}^{n} |\langle w_0^i, x_j^1 \rangle| \|x_j^1\| + \frac{\eta}{\sqrt{m}n} \sum_{i=1}^{n} |\langle w_1^i, x_j^1 \rangle| \|x_j^1\|$$

$$\leq \frac{2\eta}{\sqrt{m}n} \sum_{i=1}^{n} |\langle w_0^i, x_j^1 \rangle| \|x_j^1\| + \frac{\eta}{\sqrt{m}n} \sum_{i=1}^{n} |\langle w_1^i - w_0^i, x_j^1 \rangle| \|x_j^1\|$$

$$\lesssim \frac{2\eta}{\sqrt{m}} + \frac{\eta^2}{m} = O\left( \frac{2\eta}{\sqrt{m}} \right)$$

$$\|w_3^i - w_0^i\| \leq \frac{3\eta}{\sqrt{m}n} \sum_{i=1}^{n} |\langle w_0^i, x_j^1 \rangle| \|x_j^1\| + \frac{\eta}{\sqrt{m}n} \sum_{i=1}^{n} |\langle w_1^i - w_0^i, x_j^1 \rangle| \|x_j^1\|$$

$$+ \frac{\eta}{\sqrt{m}n} \sum_{i=1}^{n} |\langle w_2^i - w_0^i, x_j^1 \rangle| \|x_j^1\|$$

$$\lesssim \frac{3\eta}{\sqrt{m}} + \left( \frac{\eta}{\sqrt{m}} \right)^2 + \left( \frac{\eta}{\sqrt{m}} \right)^3 = O\left( \frac{3\eta}{\sqrt{m}} \right)$$

Therefore, $\|w_t^i - w_0^i\| = O\left( \frac{t\eta}{\sqrt{m}} \right)$ when $\eta = O_m(1)$. We also have

$$\|w_t - w_0\| = O(t\eta).$$

By Eq. 17:

$$|\Phi(\bar{w}_{KT}, x) - \Phi_m(w_{KT}, x)| = \widetilde{O}\left( \frac{1}{\sqrt{m}} + \frac{\eta^2 K^2 T^2}{\sqrt{m}} + \frac{\eta^2 K^2 T^2}{m} \right) = \widetilde{O}\left( \frac{\eta^2 K^2 T^2}{\sqrt{m}} \right) \quad (18)$$

Recall that we had chosen $\eta T = \Theta(d^2)$ to guarantee $\text{sign}(\Phi(\bar{w}_{KT}, x_k)) = y_k$ and $|\Phi(\bar{w}_{KT}, x_k)| \gtrsim 1$, therefore if

$$m = \widetilde{\Omega}(d^8 K^4),$$

the finite width error is small enough to conclude $\text{sign}(\Phi_m(w_{KT}, x_K)) = y_K$ for any $x_K, y_K$ from the $K$th task data distribution. Similarly, we have $\text{sign}(\Phi_m(w_{KT}, x_k)) = y_k$ for any $x_k, y_k$ from the $k$th task's data distribution because the error terms defined above are independent of the data distribution. Thus, the characterization of forgetting we derived in Eq. 15 is accurate for the same width.

Finally, we note that with the given assumptions on $n, T, m, K$ it holds that $\Phi_m(w_{KT}, x)$ is always bounded by 1. To see this, recall by Eq. 15 and Eq. 18, the network output for any training point $x$ is at most hte following:

$$\Phi_m(w_{KT}, x) \leq \eta T \left( \frac{y_k}{d^2} \pm O \left( \frac{\sqrt{K}}{d^2 \operatorname{poly} \log(d)} + \frac{\sqrt{K}}{d\sqrt{n}} \sqrt{\log(1/\delta)} \right) \right) + O \left( \frac{\sqrt{\log(1/\delta)}}{\operatorname{poly} \log(d)} \right)$$
$$+ \widetilde{O} \left( \frac{\eta^2 K^2 T^2}{\sqrt{m}} \right)$$

Recall $K = \widetilde{O}_d(1)$, with the choice of $m, n$ in the statement of the theorems, it holds with high probability that $\Phi_m(w_{KT}, x) \leq 1$. Thus, the network output always lies in the linear part of the hinge-loss for any $\eta T \leq d^2$ even at initialization where $T = 0$. Therefore, our assumption on the linearity of loss is valid throughout training.

## D  REGULARIZED CONTINUAL LEARNING: PROOF OF PROPOSITION 1

**Proposition 3** (Restatement of Prop. 1). *Consider the regularized continual learning problem Eq.6 with same setup as Theorem 1 with $m \to \infty$. The iterates of this algorithm with step-size $\eta$ are equivalent to unregularized continual learning with step-size $\widetilde{\eta}_T$ where $\widetilde{\eta}_T = \alpha_T \eta / T$ and $\alpha_T = \frac{1 - (1 - \eta\lambda)^T}{\eta\lambda}$.*

*Proof.* In regularized continual learning, the objective at task $k \geq 2$ is:

$$\min_w \widehat{F}_k(w) + \frac{\lambda}{2} \|w - w_{k-1}\|^2$$

The GD update rule is the following:

$$w_k^{(t+1)} = w_k^{(t)} - \eta \nabla \widehat{F}_k(w_k^{(t)}) - \eta\lambda(w_k^{(t)} - w_{k-1})$$
$$= (1 - \eta\lambda)w_k^{(t)} - \eta \nabla \widehat{F}_k(w_k^t) + \eta\lambda w_{k-1}.$$

For the first task, there is no regularization, therefore for neuron $i$ (we drop $i$ here for ease of notation):

$$w_1^{(1)} = w_1^{(0)} + \eta \frac{1}{\sqrt{m}} \frac{1}{n} \sum_{j=1}^n a_i \phi'(\langle w_1^{(0)}, x_j^1 \rangle) x_j^1 y_j^1$$

$$w_1 := w_2^{(0)} = w_1^{(T)} = w_1^{(0)} + \frac{\eta T}{\sqrt{m}} \frac{1}{n} \sum_{j=1}^n a_i \phi'(\langle w_1^{(0)}, x_j^1 \rangle) x_j^1 y_j^1$$

For the second task, due to the regularization term $\lambda \|w - w_1\|^2 / 2$, the first GD update takes the following shape:

$$w_2^{(1)} = (1 - \eta\lambda)w_2^{(0)} + \eta \frac{1}{\sqrt{m}} \frac{1}{n} \sum_{j=1}^n a_i \phi'(\langle w_1^{(0)}, x_j^2 \rangle) x_j^2 y_j^2 + \eta\lambda w_1$$

$$= w_2^{(0)} + \eta \frac{1}{\sqrt{m}} \frac{1}{n} \sum_{j=1}^n a_i \phi'(\langle w_1^{(0)}, x_j^2 \rangle) x_j^2 y_j^2.$$

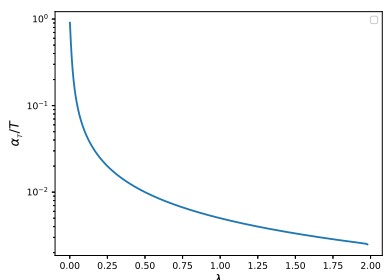 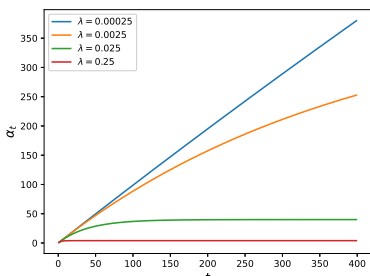

Figure 7: Effective step-size for regularized continual learning $\alpha_t$ in Prop. 1 based on regularization parameter $\lambda$ (Left) and number of GD steps $t$ (Right).

Hence, the first step is identical to the unregularized update rule. For the second step,

$$w_2^{(2)} = (1 - \eta\lambda)w_2^{(1)} + \eta\frac{1}{\sqrt{m}}\frac{1}{n}\sum_{j=1}^{n}a_i\phi'(\langle w_1^{(0)}, x_j^2\rangle)x_j^2 y_j^2 + \eta\lambda w_1$$

$$= w_2^{(0)} + ((1 - \eta\lambda) + 1)\frac{\eta}{\sqrt{m}}\frac{1}{n}\sum_{j=1}^{n}a_i\phi'(\langle w_1^{(0)}, x_j^2\rangle)x_j^2 y_j^2.$$

Similarly,

$$w_2^{(3)} = (1 - \eta\lambda)w_2^{(2)} + \eta\frac{1}{\sqrt{m}}\frac{1}{n}\sum_{j=1}^{n}a_i\phi'(\langle w_1^{(0)}, x_j^2\rangle)x_j^2 y_j^2 + \eta\lambda w_1$$

$$= w_2^{(0)} + ((1 - \eta\lambda)((1 - \eta\lambda) + 1) + 1)\frac{\eta}{\sqrt{m}}\frac{1}{n}\sum_{j=1}^{n}a_i\phi'(\langle w_1^{(0)}, x_j^2\rangle)x_j^2 y_j^2.$$

Therefore for $t \leq T$ :

$$w_2^{(t)} = w_2^{(0)} + \alpha_t\frac{\eta}{\sqrt{m}}\frac{1}{n}\sum_{j=1}^{n}a_i\phi'(\langle w_1^{(0)}, x_j^2\rangle)x_j^2 y_j^2.$$

The same steps can be repeated for every task to obtain:

$$w_k^{(t)} = w_k^{(0)} + \alpha_t\frac{\eta}{\sqrt{m}}\frac{1}{n}\sum_{j=1}^{n}a_i\phi'(\langle w_1^{(0)}, x_j^2\rangle)x_j^2 y_j^2,$$

which leads to the following expression for any $k \geq 2$ :

$$w_k := w_1^{(0)} + \frac{\eta T}{\sqrt{m}}\frac{1}{n}\sum_{j=1}^{n}a_i\phi'(\langle w_1^{(0)}, x_j^1\rangle)x_j^1 y_j^1 + \alpha_T\frac{\eta}{\sqrt{m}}\frac{1}{n}\sum_{\kappa=2}^{k}\sum_{j=1}^{n}a_i\phi'(\langle w_1^{(0)}, x_j^\kappa\rangle)x_j^\kappa y_j^\kappa$$

where

$$\alpha_1 = 1, \alpha_t = (1 - \eta\lambda)\alpha_{t-1} + 1 \quad \text{for} \quad t > 1$$

We can find the following closed form expression to the equations above: $\alpha_t = \frac{1 - (1 - \eta\lambda)^t}{\eta\lambda}$. This completes the proof. $\qquad\square$

With an accurate approximation, we have

$$\alpha_t \approx \frac{1 - e^{-\eta\lambda t}}{\eta\lambda}.$$

For small $t$, we have $\alpha_t \approx t$, whereas for large $t \approx T$, assuming $\lambda = c/T$ : we have $\alpha_t = \frac{T(1 - e^{-\eta c})}{\eta c}$. Figure 7 illustrates $\alpha_T/T$ versus regularization parameter $\lambda$ and $\alpha_t$ based on $t$ for different regularization parameters. Note that larger values of $\lambda$ correspond to smaller values of $\alpha_t$ leading to weights moving shorter distances from their initialization points. As $\lambda \to 0$, we have $\alpha_T/T \to 1$, as the step-size for regularized problem converges to the step-size for unregularized one.

## E    ADDITIONAL EXPERIMENTS AND IMPLEMENTATION DETAILS

**Experiments with MNIST and FashionMNIST.**   In Figure 11 (Top), we consider continual binary classification of digits from the MNIST dataset with $K = 2$ tasks. The plots show the amount of increase in training loss of task 1, during learning task 2. The results are averages over 15 independent experiments. For the left plot tasks are determined according to digits $0 - 3$ and for the right plot the tasks are determined according to the data distribution formed by digits $4 - 7$. The sample-size for the first task is fixed to $n = 50$ in all curves and different curves correspond to different sample sizes for the second task. The results of previous figures on the role of sample-size continue to hold for this distribution as well, since increasing the sample-size for the second task generally improves the continual learning of the first task. In Figure 11 (Bottom), we consider a similar experiment but with the FashionMNIST dataset, choose logistic loss and ReLU activation, and set the total number of tasks to $K = 4$, where different tasks correspond to data from different labels. Similar to the last experiment, we observe that increasing the sample-size for subsequent tasks generally has a positive impact on the first task's training loss.

**Experiments with transformers and GMM data.**   We also conduct experiments on attention-based architecture in Figure 12. We plot the train-time forgetting for task 1 for $K = 2$ overall tasks for a transformer with feedforward neural networks in both the encoder and the decoder parts where we consider $m_{encoder} = 60, m_{decoder} = 30$ for the left plot and $m_{encoder} = m_{decoder} = 10$ for the right plot. Results shown are averaged over 10 independent experiments. We remark that for the transformer with smaller size, we observe the similar behavior we observed for neural network experiments, i.e, increasing the sample-size for the second task can noticeably help with train-time forgetting of the first task. On the other hand, for the larger network, the behavior is more complex: increasing $n$ can help up to a certain threshold ($n \approx 250$), while above this threshold increasing $n$ hurts continual learning. While we hypothesize this behavior is due to the complex landscape of larger networks, a more thorough investigation is needed.

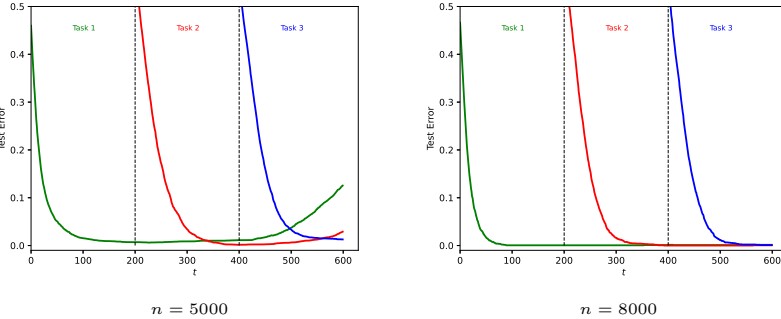

Figure 8: Classification test error for each task vs iterations for the XOR cluster with $K = 3$ tasks trained on a quadratic network with $n = 5000$(left) and $n = 8000$(right) training samples per task.

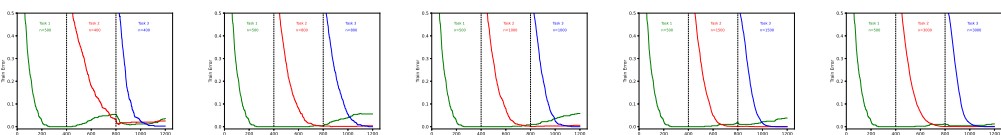

Figure 9: Repeating the experiment of Figure 3 but with ReLU activation and logistic loss.

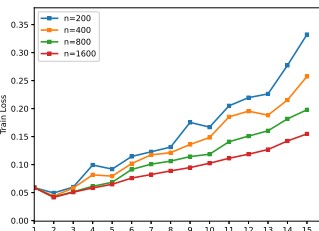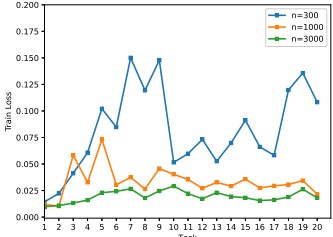

Figure 10: Train loss on task 1 as a function of the task index (i.e., $\widehat{F}_1(w_k)$ vs. $k$) for $K = 15$ and $K = 20$ tasks with $n$ samples per task for the XOR cluster dataset. The left plot uses GELU activation with logistic loss, while the right plot uses quadratic activation with hinge loss.

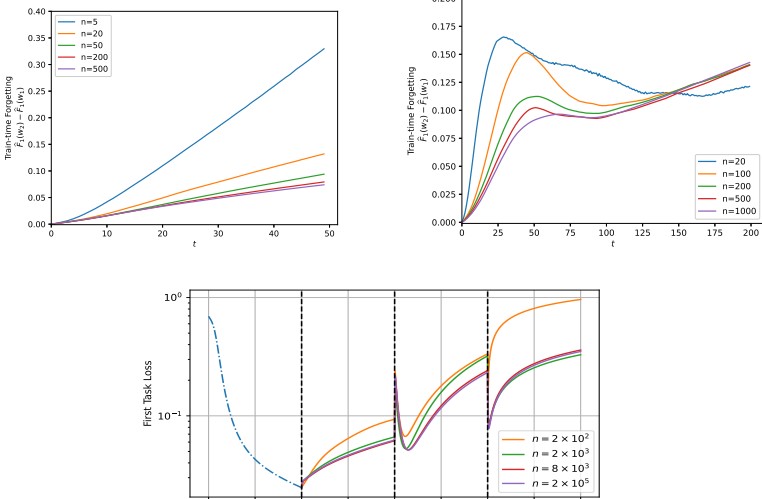

Figure 11: Top: Train-time forgetting for task 1 while learning the second task for $K = 2$ total tasks from the MNIST dataset, classifying labels '0'/'1' for task 1 and labels '2'/'3' for task 2 (left) and labels '4'/'5' for task 1 and labels '6'/'7' for task 2 (right). We fix $n = 50$ samples for the first task, and change $n$ for the second task. Bottom: First task's training loss ($\widehat{F}_1(w^{(t)})$) vs $t$ for learning 4 binary tasks from the split FashionMNIST dataset. We fix $n = 200$ for Task 1 and plot the training curves while increasing $n$ for subsequent tasks.

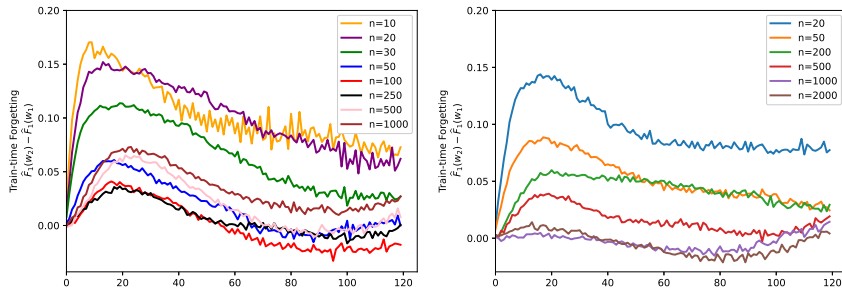

Figure 12: Train-time forgetting for task 1 based on $t$ for $K = 2$ tasks with attention-based transformers with large neural net for the encoder and decoder parts(Left plot), and with small neural net(Right plot). Here we consider a tokenized multi-task Gaussian-mixture data where the goal is to find the binary label used for each context window. We fix $n = 50$ for the first task and change $n$ for the second task. Note that our insights from previous theoretical and empirical results partially hold for this setting, especially for the transformer with smaller FFN layer.

**Implementation Details for all experiments.** We include the actual values for different problem parameters used in the numerical experiments:

Figure 1: $n = 2500$ (left), $n = 5000$(right), for both plots we set $d = 50, m = 1000, \eta = 2, T = 200, \sigma = 0.1/\sqrt{d}$ and use linear loss and quadratic activation.

Figure 2: $d = 50, m = 1000, \eta = 2, T = 200, \sigma = 0.1/\sqrt{d}$.

Figure 3: GELU activation and logistic loss. $d = 50, m = 400, \eta = 3, T = 400, \sigma = 0.1/\sqrt{d}$.

Figure 4: GELU activation, logisitc loss for both plots. We set $d = 50, m = 2000, \eta = 30, T = 2000, \sigma = 0.2/\sqrt{d}$. Right: $n = 2000, T = 4000$.

Figure 5: We set $n = 5000, d = 75, \eta = 5, T = 200, \sigma = 0.15/\sqrt{d}$ and vary $m = 100, 300, 1000, 3000, 6000, 10000$.

Figure 6: GELU activation, Logistic loss, $d = 50, n = 200, \eta = 20, T = 1000, \sigma = 0.2/\sqrt{d}$

Figure 8: $n = 5000$(left),8000(right),$d = 75, m = 1000, \eta = 5, T = 200, \sigma = 0.15/\sqrt{d}$, linear loss, quadratic activation

Figure9: Using the same setup as Figure 3 but with ReLU activation and logistic loss. $d = 50, m = 1000, \eta = 0.3, T = 400, \sigma = 0.1/\sqrt{d}$

Figure 10: GELU activation and logisitc loss, $\eta = 30, m = 400$ for the left plot, Quadratic activation and Hinge loss, $m = 1000, \eta = 4$ for the right plot. For both plots we set, $d = 50, T = 400, \sigma = 0.1/\sqrt{d}$.

Figure 11: Top: $n = 50$ samples for the first task, $n$ varying for the second task, GELU activation, Hinge loss, $d = 784, m = 500$. For the left plot $\eta = 0.0003, T = 50$ and for the right $\eta = 0.001, T = 200$. The results are averages over 15 experiments. Bottom: $T = 2000, \eta = 0.05, m = 2000, K = 4$, ReLU activation and Logistic loss, Tasks are chosen from labels 1-4, 7-10 from the FMNIST dataset. Dataset is normalized to have $\ell_2$-norm at most 1.

Figure 12: We use hinge-loss, ReLU activation, and the transformer is one-layer with one head, context length = 10, the hidden-layer size of the feedforward neural is 60 and for the decoder is 30. In the right plot, both hidden-layer sizes are reduced to 10 $\sigma = 0.1/\sqrt{d}, \mu^k = \mathbf{e}_k/\sqrt{d}$ for $k \in [2]$, $\eta = 0.01$.