# OpenReview forum: "On the Theory of Continual Learning with Gradient Descent for Neural Networks"
_ICLR.cc/2026/Conference — Submitted to ICLR 2026_

### Official Review · Reviewer_v8Eo · 2025-10-26

**Soundness:** 2
**Presentation:** 3
**Contribution:** 2
**Rating:** 6
**Confidence:** 3

**Summary:**

This paper presents a theoretical analysis of continual learning (CL) when using standard gradient descent (GD) on neural networks. To make the problem tractable, the authors focus on a specific, stylized setting: a one-hidden-layer quadratic neural network trained on a sequence of tasks drawn from an XOR cluster dataset with orthogonal means. The main contribution is the derivation of closed-form bounds on both train-time and test-time forgetting. These bounds explicitly characterize the rate of forgetting in terms of key problem parameters: the number of iterations ($T$), the sample size ($n$), the number of tasks ($K$), and the hidden-layer size ($m$). The paper's numerical experiments on this synthetic data (and a small MNIST experiment in the appendix) serve to confirm the derived theoretical relationships.

**Strengths:**

1. The paper is well-motivated by the general lack of deep theoretical understanding of how standard GD behaves in a continual learning setting. Choosing a simplified, tractable model (quadratic network on XOR data) to gain analytical insight is a valid and valuable research methodology.

2. The primary strength is the derivation of explicit, closed-form bounds (e.g., in Theorem 1) that connect forgetting to $n$, $m$, $T$, and $K$. This provides a concrete, analytical basis for understanding the trade-offs that are often observed empirically (e.g., the roles of overparameterization and sample size in mitigating forgetting).

**Weaknesses:**

1. My main reservation is the highly specific nature of the theoretical setup. The results are derived for one-hidden-layer quadratic networks on XOR cluster data with orthogonal means. It is very difficult to ascertain how—or even if—these specific bounds and polynomial dependencies (e.g., $m=\tilde{\Omega}(d^8K^4)$) would generalize to the settings where CL is practically studied: multi-layer ReLU networks, general classification/regression problems, and non-orthogonal tasks.

2. The experiments are not sufficient to bridge the gap between this specific theory and general practice. The main experiments are on the same synthetic XOR data used for the theory, which only confirms the calculations. The appendix includes an experiment on MNIST, but it appears to use only two tasks. A two-task sequence is not a compelling demonstration for continual learning, which is concerned with performance over a sequence of many tasks.

3. The paper would be significantly stronger if it demonstrated that the qualitative insights from its theory hold on more complex, standard CL benchmarks (e.g., Split CIFAR-10 or Split CIFAR-100) with a more realistic number of tasks (e.g., 5, 10, or more). This would provide evidence that the core relationships identified (e.g., between forgetting and $n$ or $m$) are fundamental and not just artifacts of the specific XOR setup.

**Questions:**

1. Could the authors provide some intuition on how they expect these results to change for different, non-quadratic activation functions like ReLU? How much of the analysis is critically dependent on the quadratic activation and the specific XOR cluster structure?

2. The two-task MNIST experiment in the appendix is quite limited. Could the authors provide further empirical validation on a standard benchmark like Split CIFAR-10 with 5 or 10 tasks to demonstrate that the qualitative trends predicted by the theory (e.g., the scaling of forgetting with $n$, $m$, and $K$) hold in a more complex and practical setting?

---

> ### Author Response · Authors · 2025-11-21
>
> Thanks for your time and review. Please find our response below, and we'd be glad to answer your further questions.
>
>
> Extensions to ReLU or other similar data distributions for train-time forgetting are within reach. To see this note that the final expression we have for weights holds for any activation function or data distributions. In particular, Eq. in line 1003 in the appendix holds for virtually any activation function and data distribution. The relevant calculations for forgetting bounds, up to Eq. 14 also hold for any data distribution and the quadratic activation. The remaining steps are to calculate the model output for the given weights. Here we assume a distribution for data that is necessary for obtaining tight concentration results and characterizing forgetting. Extensions to other data distributions would require computing Eq. 14. based on the distribution’s assumptions. We expect the analysis to extend to other clustered data distributions with sub-Gaussian noise within the NTK regime.  The calculations on finite-width error and the impact of $m$ in section C.2. are independent of data distribution, therefore it holds for any other data distributions with bounded norm.
>
> For the generalization analysis, while the claims hold for almost any smooth activation and data distribution, they crucially depend on the smoothness assumption.  Extensions to ReLU are interesting future directions. We expect these extensions do not qualitatively change the results as shown by the experiments (fig. 9).
>
>
> Regarding experiments with more complex datasets, note that training CIFAR 10 in the kernel regime (that we study in the paper) is computationally inefficient. In fact it would require choosing a small learning rate and massively over-parameterized deep neural nets.  However, we have now included a figure in the revision (Fig 11-bottom) on train-time forgetting during learning 4 tasks from the FashionMNIST dataset with a  two-layer neural net (which is close to our paper’s setup).

---

> > ### Comment · Reviewer_v8Eo · 2025-11-21
> >
> > Thank you for the response. I understand that it may be difficult to extend to multi-layer networks. But I believe that the paper would benefit if the authors could include at least some discussion or the insight on how to bridge the theory to the practice. Meanwhile, I am still wondering how the specific bounds would generalize to general classification/regression problems and non-orthogonal tasks. Why and where is the XOR cluster data with orthogonal means necessary in the analysis?

---

> > > ### Author Response · Authors · 2025-11-22
> > >
> > > Thank you for the follow-up question. Assumptions on the data distribution or on the training regime (such as the NTK regime) are necessary for obtaining closed-form bounds and for deriving tight sample/iteration complexities. Current theoretical analyses of neural networks with provable guarantees are largely limited to multi-index data models, and the XOR-cluster model serves as a canonical example within this class.
> > > Importantly, the proof techniques underlying Theorems 1–2 (up to the steps involving the explicit model-output expression in Eq. 14 and the forgetting equation in Eq. 16) apply to a broad family of data distributions. These steps do not rely on any special geometric structure beyond concentration of the empirical NTK around its population counterpart and uniform boundedness of the inputs. The parts of the argument that do specialize to the XOR-cluster distribution arise only when we derive explicit closed-form formulas for the network output and when we obtain closed-form expressions for forgetting.
> > > Generalizing these explicit formulas to other data distributions (e.g., other clustered or multi-index models) would require replacing the closed-form computations with distribution-specific estimates. We expect the qualitative dependencies on
> > > $n,m,T,\eta,K$ to remain similar for other sub-Gaussian clustered task distributions, although the exact constants and forms of the bounds will differ. We have added a remark clarifying this point in the updated manuscript (highlighted in red after Theorem 2).
> > >
> > > Regarding the assumption of orthogonal task means: this setting captures the case in which tasks are uncorrelated, for example, when the means are drawn uniformly at random from a sphere. For general non-orthogonal tasks, the same analytical framework still applies, but the correlation constants between task means enters the explicit calculations. The resulting approach remains valid but no longer admit the simple closed forms obtainable under orthogonality.

---

### Official Review · Reviewer_8BnG · 2025-10-30

**Soundness:** 2
**Presentation:** 3
**Contribution:** 2
**Rating:** 2
**Confidence:** 4

**Summary:**

The paper analyzes continual learning with gradient descent on a two-layer quadratic network trained on a multi-task XOR-cluster distribution (orthogonal task means, Gaussian noise). It shows that test-time forgetting for task $k$ after learning subsequent tasks can be decomposed into (i) train-time forgetting (an increase in empirical loss) and (ii) a delayed generalization gap.

**Strengths:**

1. The paper is well written and well structured.

**Weaknesses:**

1. The motivation is not clearly or strongly articulated.
2. The theoretical results are confusing. In Theorem 1, the logarithmic factors cannot be omitted, because without them it is unclear how the final bounds decay with the dimension $d$. Given the stated conditions on $\eta T, n$, and $m$, the dependency on $d$ remains unclear.
3. It is not clear how Theorem 4 improves upon Theorem 3. Under the assumption of sufficiently large model width $m$, the bound is improved, and an example is given (line 311), but this seems unfair: the paper should explain the width assumption in more detail and also report the corresponding Theorem 3 results under the same width condition for a fair comparison.

**Questions:**

Above

---

> ### Author Response · Authors · 2025-11-21
>
> Thanks for your time and review. We have addressed your concerns below and we'd be glad to answer your further questions.
>
>
> 1- *The motivation is not clearly or strongly articulated.*
>
> The theoretical understanding of continual learning in literature is very limited.. Theoretical studies of neural networks’ behavior  with single- or multi-index models have attracted significant attention in recent years (see sec. 2.1.2 in the paper) but were limited to stationary single-task settings. Our work gives the first closed-form bound for continual learning for neural nets and it is a stepping stone for a principled study of catastrophic forgetting in neural networks.
>
> 2- *The theoretical results are confusing. In Theorem 1, the logarithmic factors cannot be omitted,*
>
> Given the conditions $\eta T = d^2$ and $n= d^2*log^c(d)$ for some sufficiently large $c$ and noting that $K=O(log(d))$, the  logarithmic terms cancel, and with the poly-log term which is $log^c(d)$ for some sufficiently large constant c, the bound in Theorem 1 decays with the rate $O(1/polylog(d)).$
>
> 3-*It is not clear how Theorem 4 improves upon Theorem 3*
>
> The width lower-bound condition in Thm 4 is “smaller” than Thm 3.  To see this note that, for Thm 3 to lead to meaningful bounds it should hold that $m> T^2 * K^2$ which has a larger dependence on T compared to Thm 4 which is polylogarithmic in $T$. The bound discussed in Remark 2 also emphasizes that it remains valid even if $T$ is large. Note that the per task loss converges to zero, thus we expect $\sum_{t=0}^{T-1} \widehat{F}_j\left(w_j^{(t)}\right)$ to be much smaller than $T$ (in fact logarithmic in $T$). Therefore, the bound in Thm4 improves both the width condition and the final bound.
>
> Finally,  as discussed in the paper, we would like to emphasize that Thm 3 is sufficient for obtaining our desired results and the improvements on Thm4 only serve as an improvement to show that the dependence on $T$ can be mitigated.

---

### Official Review · Reviewer_XcUj · 2025-10-31

**Soundness:** 3
**Presentation:** 3
**Contribution:** 3
**Rating:** 4
**Confidence:** 3

**Summary:**

This paper provides a theoretical analysis of continual learning in neural networks trained via gradient descent. Focusing on a simplified setting: a one-hidden-layer quadratic network trained on an XOR cluster dataset with Gaussian noise. The authors derive quantitative bounds on the rate of forgetting across tasks. They show how factors such as the number of iterations, sample size, task count, and hidden-layer width influence forgetting during both training and testing. The theoretical findings are supported by experiments on various setups, suggesting that the identified mechanisms generalize beyond the specific analytical model.

**Strengths:**

The paper tackles an important and timely problem: understanding the theoretical performance of continual learning in neural networks. It is well structured, combining rigorous theoretical analysis with supporting numerical experiments that validate the findings.

**Weaknesses:**

Some aspects of the system setup require stronger justification. In particular, the neural network model appears overly simplified and departs from architectures typically used in practice, which may limit the generality of the conclusions.

**Questions:**

1. In the first equation of Section 2.1.1, the function $f$ is not clearly defined. Please specify what types of functions are admissible for $f$, and clarify why its input involves $y_i$ multiplied by $\Phi$.
2. Is the quadratic activation used in prior works? Was it chosen purely for mathematical convenience in analysis, or does it have theoretical or empirical motivation?
3. The dimensions of $x$ and $y$ are not specified when first introduced in Section 2.1.1. Based on Eq. (1), it appears $x$ is a $d$-dimensional real vector and $y$ is binary ($\pm 1$).
4. The explanation of Eq. (3) is too brief. The paper states that "in the interpolating regime where the network can achieve zero training loss, we can drop the last term," but the explicit form of the training loss is not provided (related to my Question 1). It is unclear whether the loss is always nonnegative and whether omitting this term could lead to a loose approximation.
5. The neural network model omits bias terms, i.e., it uses $x^T w_i$ instead of $x^T w_i + b_i$ as input to the activation function. Please justify this modeling choice and discuss its implications for generality.

---

> ### Author Response · Authors · 2025-11-21
>
> Thanks for your time and review. We have addressed your concerns below and we'd be glad to answer further questions.
>
>
> 1-*Please specify what types of functions are admissible for , and clarify why its input involves $y$  multiplied by $\Phi$* .
>
> The function $f$ considered in Thm1-3 is the hinge-loss as mentioned in line 232. The fact that the label $y$ is multiplied by the model output $\Phi$ is the common choice for binary classification tasks in both ML theory and practice(see prior works such as [ji and telgarsky 2020, glasgow 2023, nitanda et al 2019, taheri et al 2024, telgarsky 2023])
>
> 2- *Is the quadratic activation used in prior works? Was it chosen purely for mathematical convenience in analysis, or does it have theoretical or empirical motivation?*
>
> Thms 3-4 hold for almost any smooth activation function.  Therefore, we need the smoothness assumption in order to combine the results from thms1-2 with thm3-4. The quadratic activation is considered in some prior works (e.g. Ghorbani et al 2019, Taheri et al 2024). While the choice of quadratic activation is especially motivated given the quadratic nature of the data model, we expect using other activation functions is also possible without dramatically changing the analysis. We also highlight that (as our experiments show) our results are insightful for different activations and losses.
>
> 3- *The dimensions of  and  are not specified when first introduced in Section 2.1.1.*
>
> Yes, $x$ is d-dimensional and $y \in { \pm 1}$. We have updated the paper to better reflect this.
>
> 4- *The explanation of Eq. (3) is too brief.*
>
> The loss function we consider (hinge-loss) is non-negative (rf. Sec. 2.2, also see the updated section 2.1.1). Omitting the last-term does not make the bound loose. In particular, the last-term can be bounded by the per-task generalization error, which is sufficiently small given our assumptions on $n,T,m$.
>
> 5-*The neural network model omits bias terms*
>
> Our goal was to gain insights based on simple popular methods. Our analysis can be extended to the Include bias terms. However, in-line with the majority of prior theoretical works (see references above) in literature, we do not believe bias terms are needed for the studied problem in this paper.

---

### Official Review · Reviewer_i38r · 2025-11-02

**Soundness:** 2
**Presentation:** 3
**Contribution:** 1
**Rating:** 2
**Confidence:** 5

**Summary:**

```
This paper studies the generalization ability of two-layer neural networks (NNs) trained from continual learning (CL). The authors derive one upper bound for the CL training loss (Theorem 1) and two upper bounds for the CL generalization errors (Theorem 3 and Theorem 4). They then conduct experiments on both synthetic and real-world data to analyze their theoretical findings in practice.
```

**Strengths:**

```
The paper itself is well-written and easy to follow.
```

**Weaknesses:**

```
1. The proved theorems in this paper seem to give no novel insight into the studies of CL, such as how to efficiently improve the performance of CL models and how to effectively avoid catastrophic forgetting. For example, from Theorem 3, while one can only know that increasing the number of training samples for each task helps improve the generalizability of models trained from CL, but "more training data helps generalization" is actually a trivial conclusion.

2. Both the proved Theorem 3 and Theorem 4 do not seem technically strong enough.
    - For Theorem 3, the generalization upper bound has a positive correlation with $T$, i.e., the number of SGD iterations for each task in CL. This means that performing more SGD actually hurts the generalizability of CL models, which is a very weird conclusion and is actually contradictory to the "benign overfitting phenomenon".
    - The authors then improve Theorem 3 to Theorem 4 so that the CL generalization upper bound no longer explicitly depends on the SGD iteration number $T$. However, this is achieved by making some very weird and unrealistic assumptions (see the new conditions in Theorem 4). In particular, I think it is inappropriate to assume that $m$ is greater than some training-dependent terms $\hat F_j(w^{(t)}_j)$, especially given that the parameter $w^{(t)}_j$ also depends on $m$ ($w^{(t)}_j$ is a real vector of length $m\cdot d$ according to Section 2.1.1). Furthermore, even with those overly strong assumptions, the improved generalization upper bound in Eq.(5) is still a sum of $T$ terms, which means it still implicitly depends on the value $T$.

3. The authors only proved upper bounds for the generalization error, but not lower bounds. As a result, one cannot tell whether the upper bounds proved in this paper are tight or not, which shrinks the contribution of this paper.

4. A vast body of existing literature has studied CL with NNs, such as [r1, r2, r3, r4]. In particular, [r1] also focuses on studying two-layer NNs. So I think the authors should make an in-depth comparison of how their results differ from existing theoretical works on CL with NNs and what the advantages of their proved generalization upper bounds are.


**References**

[r1] Li et al. Towards Understanding Catastrophic Forgetting in Two-layer Convolutional Neural Networks. ICML 2025.

[r2] Benjamin et al. Continual learning with the neural tangent ensemble. NeurIPS 2024.

[r3] Andle et al. Theoretical understanding of the information flow on continual learning performance. ECCV 2022.

[r4] Cao et al. Provable lifelong learning of representations. AISTATS 2022.
```

**Questions:**

```
See **Weaknesses**.
```

---

> ### Author Response · Authors · 2025-11-21
>
> Thanks for your time and review. We have addressed your concerns below.
>
> 1-*"The proved theorems in this paper seem to give no novel insight into the studies of CL…"more training data helps generalization" is actually a trivial conclusion."*
>
> Our findings are insightful for understanding continual learning in the kernel regime of neural networks’ training  and to the best of our knowledge, they are novel. In what follows we clarify the new results obtained from our derivations and we would be glad to answer any further questions.
>
> Regarding the role of sample size ($n$), our theoretical results (Thm 1-4) show given large n and sufficient over-parameterization, the forgetting in *both training time and test time* is asymptotically *zero*. We find this result surprising and unique to the kernel regime training of neural nets, due to the linearization of the model in the kernel regime. Note that the statement “increasing sample-size helps” does not necessarily always hold, for example see the left panel in Figure 12 for learning GMMs with transformers, where increasing the sample-size of task 2 after a threshold increases the forgetting of task 1.
> To the best of our knowledge, none of the above results are discussed in literature, and it remains unclear whether they hold for other setups such as the feature learning regime with neural networks or for other models such as transformers, which are left for future work.
> We also showed in prop. 1 that the regularized continual learning, which is a popular continual learning algorithm, can be studied within our framework, therefore the bounds remain valid even for continual learning with regularization.
>
> 2.*”Both the proved Theorem 3 and Theorem 4 do not seem technically strong enough”*.
>
> We believe the reviewer may have some misunderstanding or unfamiliarity with the prior theoretical works on neural net’s optimization and generalization. Please note that almost all bounds based on algorithmic stability or Rademacher complexity for neural networks and convex objectives in literature have, either explicitly or implicitly, a dependence on “T” [see references [1-6]], whether using SGD or GD (we’re using GD throughout the paper). In fact, these bounds are known to be tight for standard setups (see reference [7]). These results are not contradictory to the benign overfitting phenomenon: first our setup/results do not apply to the benign-overfitting setting, as analyzing it would generally require specific assumptions such as label-noise and avoiding early-stopping in order to allow memorization of noisy training samples. Second, note that Theorems 3-4 hold for any data distribution and therefore early stopping is necessary.
> The dependence on summation of training loss for T epochs, is very mild on $T$ given that with the common training-rate of $\tilde O(1/t)$ at iteration $t$, the bound would transform to a poly-logarithmic term. Please see our Remark 2.
>
> The conditions on width are also natural for the NTK setup. In fact, for separable data with positive NTK-margin $\gamma$, it is known that the distance travelled by weights is constant with respect to $m$. See [Ji and Telgarsky 2020, Taheri et al 2024].
>
>
> All that being said, we would like to emphasize that Thm3 is sufficient for obtaining our desired results and the improvements on Thm4 only serve as an improvement to show that the dependence on $T$ can be mitigated.
>
>
> 3.*"The authors only proved upper bounds for the generalization error, but not lower bounds"*.
>
> In our  “delayed generalization error bounds” (Thms 3-4), we have no assumption on data and they hold for almost any data distribution. Proving a lower-bound on generalization error would require constructions which are beyond the scope of this paper. Again, we remark that for the standard setups the algorithmic-stability bounds are optimal.
> Moreover, as a result of Theorems 2 and 3, we showed  the sample complexity of $n=\tilde\Theta(d^2 K)$ is sufficient for continual learning of XOR datasets. Given the quadratic nature of the XOR data distribution, prior works(e.g., see reference [8]) indicate the same lower-bound  for kernel methods in standard settings i.e. $n=\tilde\Omega(d^2)$, therefore we expect the sample-complexity to be tight up to logarithmic factors, and furthermore we show that it is sufficient for continual learning. Finally, we emphasize that our experiments verify the qualitative effects of different problem parameters.
>
> 4-*"A vast body of existing literature has studied CL with NNs, such as [r1, r2, r3, r4]..."*
>
> We have included these works in the revision. However, these works focus on different settings than ours. We note that our results are the first closed-form bounds on train/test error for continual learning with neural networks and a multi-index model, while previous known results of this flavor (e.g., (Evron et al 2023)) were limited to linear models.

---

> > ### Author Response · Authors · 2025-11-21
> >
> > References:
> >
> > [1] Fine-Grained Analysis of Stability and Generalization for Stochastic Gradient Descent, Lei, Yiming, ICML 2020
> >
> > [2] A PAC-Bayesian Approach to Spectrally-Normalized Margin Bounds for Neural Networks Neyshabur, Bhojamapali, Srebro, ICLR 2018.
> >
> > [3] Spectrally-normalized margin bounds for neural networks. Bartlett, Foster, Telgarsky, NIPS 2017.
> >
> > [4] Stability vs implicit bias of gradient methods on separable data and beyond. Schliserman and Koren, COLT 2022.
> >
> > [5] Stability & generalisation of gradient descent for shallow neural networks without the neural tangent kernel. Richards and Kuzborskij, NeurIPS 2021.
> >
> > [6] Beyond Lipschitz: Sharp Generalization and Excess Risk Bounds for Full-Batch GD. Nikolakakis, Haddadpour, Karbasi, Kalogerias, ICLR 2023
> >
> > [7] Select without Fear: Almost All Minibatch Schedules Generalize Optimally, Nikolakakis, Karbasi, Kalogerias, SIMODS, 2025.
> >
> > [8] Linearized two-layers neural networks in high dimension. Ghorbani, Mei, Misiakiewicz, and Montanari. Annals of Statistics 2021.

---

### Author Response · Authors · 2025-12-03

We thank all reviewers for their insightful review and for the opportunity to clarify several points. Below we summarize the key points of the discussion.

1- Insights of our theoretical results:
In response to Reviewer i38r, we clarified that our work provides, to the best of our knowledge, the first closed-form bounds on both train-time and test-time forgetting for neural networks in a multi-task continual learning setting. Our analysis shows that in the NTK regime, with sufficiently large sample size and over-parameterization, both forms of forgetting converge to zero, highlighting a concrete benefit of NTK-type training for continual learning. The explicit dependencies we derive on key problem parameters are new for the NTK setting and do not follow from prior continual learning theory. While theoretical studies of neural networks have grown rapidly for single-task setups, a principled analysis of continual learning for multi-index models has been missing, and our work fills this gap.

2- Lower bounds:
In response to reviewer i38r, We explained that proving distribution-free lower bounds is outside the scope of our paper. However, for the XOR multi-index model our upper bounds match known tight results for kernel methods up to logarithmic factors.

3- Relation to prior works:
The referenced works (Li et al. 2025; Benjamin et al. 2024; Andle et al. 2022; Cao et al. 2022) focus on substantially different settings, e.g., convolutional architectures, or linear models. None provide closed-form forgetting bounds in a multi-index neural network model. We included explicit comparisons in the revised version.

4- Statement of theorems:
In response to Reviewers 8BnG (and i38r), we clarified that the statement of Thm 1 is correct. Moreover, Thm 4 indeed improves the dependencies based on $T$ compared to Thm 3. However, Thm 3 is already sufficient for our theoretical claims to be valid.

5- Modeling choices and assumptions:
In response to reviewers v8EO and XcUj, We clarified the specific choices of activation functions, data distribution, orthogonality of task means, and smoothness. Many parts of our proof hold for general smooth activations and broader clustered data distributions. Orthogonality is used only to obtain closed-form expressions; the framework extends with modified constants for non-orthogonal tasks.

Moreover, in response to reviewers v8EO, We added empirical results on multi-task experiments on FashionMNIST. We also clarified that we expect our findings to hold for the kernel regime and across different relevant distributions.

We believe our response addresses the main points raised by the reviewers. We thank the reviewers and Area chair again for their thoughtful feedback and handling our submission.

---

### Meta-Review · Area_Chair_brpy · 2026-01-07

**Summary:**

This work theoretically analyzes continual learning of a solvable quadratic neural network with a specific data structure, i.e., XOR clusters. While it has the strength that the obtained results are concrete and rigorous, some reviewers pointed out the weakness that the insight into real CL research is limited.

Certainly, I think that the current theoretical setting and the obtained closed-form expressions are novel. However, to convince these reviewers, it would be necessary to enrich not the novelty of the formulation, but rather to clarify ”the advantages of the proved generalization upper bounds” compared to the previous theory of CL in other formulations. It would also be effective, to reach the bar of ICLR, to provide some insight into ”s tandard CL benchmarks (e.g., Split CIFAR-10 or Split CIFAR-100) with a more realistic number of tasks (e.g., 5, 10, or more), where, even if the model setting goes beyond the theoretical scope, the validity of the theoretical results could be qualitatively examined empirically.

Thus, I evaluate this work as a rejection.

**Reviewer Concerns:**

The following concerns remain outstanding:

**Reviewer i38r**
-  The proved theorems in this paper seem to give no novel insight into the studies of CL, such as how to efficiently improve the performance of CL models and how to effectively avoid catastrophic forgetting

- I think the authors should make an in-depth comparison of how their results differ from existing theoretical works on CL with NNs and what the advantages of their proved generalization upper bounds are.

**Reviewer XcUj**

- the neural network model appears overly simplified and departs from architectures typically used in practice, which may limit the generality of the conclusions.

**Reviewer v8Eo**

- ... qualitative insights from its theory hold on more complex, standard CL benchmarks (e.g., Split CIFAR-10 or Split CIFAR-100) with a more realistic number of tasks (e.g., 5, 10, or more).

**Reviewer Scores:**

The reviewers are fairly confident on their claims and I expect no score changes.

---

### Decision · Program_Chairs · 2026-01-26

Reject